# Recent Advances in Greener Asymmetric Organocatalysis Using Bio-Based Solvents

Lorena S. R. Martelli [ID], Ingrid V. Machado [ID], Jhonathan R. N. dos Santos [ID] and Arlene G. Corrêa *[ID]

Centre of Excellence for Research in Sustainable Chemistry, Department of Chemistry,
Federal University of São Carlos, São Carlos 13565-905, SP, Brazil
* Correspondence: agcorrea@ufscar.br

**Abstract:** Efficient synthetic methods that avoid the extensive use of hazardous reagents and solvents, as well as harsh reaction conditions, have become paramount in the field of organic synthesis. Organocatalysis is notably one of the best tools in building chemical bonds between carbons and carbon-heteroatoms; however, most examples still employ toxic volatile organic solvents. Although a portfolio of greener solvents is now commercially available, only ethyl alcohol, ethyl acetate, 2-methyltetrahydrofuran, supercritical carbon dioxide, ethyl lactate, and diethyl carbonate have been explored with chiral organocatalysts. In this review, the application of these bio-based solvents in asymmetric organocatalytic methods reported in the last decade is discussed, highlighting the proposed mechanism pathway for the transformations.

**Keywords:** bio-based solvents; asymmetric organocatalysis; sustainability; green synthesis

## 1. Introduction

Organic compounds have been used for decades as catalysts, but their application in asymmetric catalysis has become even more important in recent years due to their effective selectivity in a wide range of reactions [1]. In general, the organocatalysts are promptly available, either from natural or synthetic starting materials, therefore, saving cost, time, and energy [2]. A plethora of highly effective small-molecule organocatalysts have enriched the field of organic synthesis, including chiral proline derivatives, *N*-heterocyclic carbenes, thioureas, Brønsted acids, and phase-transfer catalysts (PTC), such as the quaternary ammonium salts derived from cinchona alkaloids [3].

Furthermore, the experimental application requires milder conditions when compared with most metal catalysts, and the low toxicity of the organocatalysts bring great utility to medicinal chemistry [4–6].

Due to these properties, organocatalysis has become an important tool for rapid and efficient building up of C-C and C-heteroatom chemical bonds [7,8]. In 2021, asymmetric organocatalysis was recognized with the Nobel Prize in Chemistry for Benjamin List [9] and David MacMillan [10], pioneers in the use of chiral secondary amines as organocatalysts and their broad understanding of asymmetric catalytic systems through enamine and iminium ions as key intermediates. Nevertheless, many other catalytic processes, volatile organic solvents, especially dichloromethane and chloroform, have been used, and relatively high catalyst loading is required, representing major drawbacks for organocatalysis [11].

Efficient synthetic methods with eco-friendliness and sustainability that avoid the extensive use of toxic and hazardous reagents and solvents, as well as harsh reaction conditions, have become paramount in the field of organic synthesis [12]. A major source of waste in chemical manufacturing, particularly in the pharmaceutical industry, is solvent losses. An inventory of waste produced by pharmaceutical manufacturers revealed that solvents and water accounted for 58% and 28%, respectively, of the residue, compared to 8% for the raw materials [13].

Waste prevention can be achieved if most of the reagents and solvents are recyclable. Using appropriate catalysts and non-volatile solvents, the reaction medium can be recycled, leading to low E factors [14]. Alternative solvents suitable for green chemistry are those that have low toxicity, are easy to recycle, inert, and do not contaminate the product [15].

Organocatalysis is continuously adapted to current chemical transformations, which in some cases allowed for overcoming the limitations [16], as well as improved enantioselective catalytic properties. Therefore, the scientific community is focused on the replacement of traditional organic solvents with greener ones in organocatalytic approaches [17], as for example, the use of eutectic mixtures and water in asymmetric organocatalysis [5,18]. Recently, Miele and colleagues reviewed the use of bio-based solvents in the asymmetric metal-, bio-, and organocatalysis [19]. Herein, we have reviewed the advances reported in the last decade on the use of bio-based solvents in asymmetric organocatalytic processes, highlighting the proposed mechanism key step of the cited examples.

## 2. Bio-Based Solvents

Solvents used in the chemical industry are often derived from non-renewable fossil resources, which are commonly harmful to nature and human health. Accordingly, there is a continuous requirement to replace them in chemical transformations with solvents derived from renewable resources, such as biomass [20]. Nevertheless, a disadvantage of bio-based solvents is related to their synthetic route, considering that in some cases, it is still necessary to improve the production cost, as well as selectivity, recycling, and yields [21].

Bio-based solvents are derived from vegetable, animal, or mineral raw materials using chemical and physical processes that are, in general, safer for the environment and humans. Due to the regulatory constraints that restrict the use of a large number of traditional volatile organic compounds (VOCs), such as halogenated or aromatic derivatives, the bio-based solvents are on the rise [22–24].

Examples of chemical reactions where the replacement of traditional solvents by bio-based solvents providing greater selectivity of the applied catalysts and an increase in reaction yields were reported. In addition, in general, these solvents show higher stability for handling when compared with the VOCs [25].

Given that their conviction is compatible with sustainable development, it is assumed that bio-solvents provide an environmental balance with reduced VOCs, biodegradability, increased safety, and non-ecotoxicity. The processes involved in solvent production must be eco-compatible, with raw material widely available at an affordable cost, as well as recyclable and easy to purify. Moreover, low toxicity and performance similar to traditional solvents are idealized [26].

In this sense, there is considerable interest in conducting organocatalyzed reactions in aqueous or semi-aqueous media due to their green nature, as well as the special solvent effects of water on organic reactions, but it is still limited to a group of reactions [27]. In the following sections, the application of the bio-based solvents ethyl alcohol, 2-methyltetrahydrofuran, ethyl acetate, ethyl lactate, supercritical carbon dioxide, and diethyl carbonate in asymmetric organocatalytic methods is discussed.

### 2.1. Ethyl Alcohol

The production of ethyl alcohol from renewable sources has been gaining momentum in recent decades, as this process does not generate toxic by-products and provides an economically viable and sustainable destination for organic matter waste [28]. Bioethanol is usually obtained from the fermentation of carbohydrates, which are highly abundant and relatively inexpensive [21]. Moreover, bioethanol can also be produced from lignocelluloses of agricultural residues, although in these processes, a pre-treatment step of the raw material is necessary due to the presence of a high amount of fibers [29].

Among the bio-based solvents, ethanol is the most common, being also used in the synthesis of other bio-based solvents, as well as in several chemical transformations, including asymmetric organocatalysis [24,30].

### 2.1.1. Catalysis Via Covalent Bonding

Chiral secondary amines have been applied as chiral organocatalysts since the beginning of asymmetric catalysis, and their activation mode occurs through the formation of a covalent bonding between the organocatalyst and the carbonyl substrate. As a result, an enamine intermediate or iminium ion is formed, thus, modifying the energy of the frontier orbitals, HOMO of the enamine and LUMO of the iminium ion. The increase in HOMO and the decrease in LUMO energies result in higher reactivity of both intermediates, so they become more nucleo- and electrophilic, respectively. The stereochemistry of products in organocatalyzed reactions by proline derivatives are determined by the presence of an acid group or steric hindrance. The reactant approach to the intermediate can occur on the same side, due to the hydrogen bonding with the acid, or on the opposite side of the bulky group [31].

The asymmetric Michael addition reaction of aldehydes **1** to nitroolefins **2** was first reported in 2005 by Hayashi, using toluene and (*R*)-diphenylprolinol trimethyl silyl ether (Jørgensen–Hayashi catalyst, 5 mol%) at room temperature [32]. In 2012, Wang and colleagues [33] reported the synthesis of the same compounds **4** employing a proline derivative **3** as a heterogeneous catalyst, in a mixture of EtOH-H$_2$O (1:1) as a solvent, with good to excellent yields and excellent enantiomeric excesses (*ee*), as well as high diastereoisomeric ratios (*dr*). Although the obtained results were slightly inferior to Hayashi's work, they have turned this transformation more sustainable, since organocatalyst **3** could be reused for four cycles without losing enantioselectivity. The choice of EtOH-H$_2$O was due to the fact that polymer **3** was insoluble in the other solvents tested (Scheme 1).

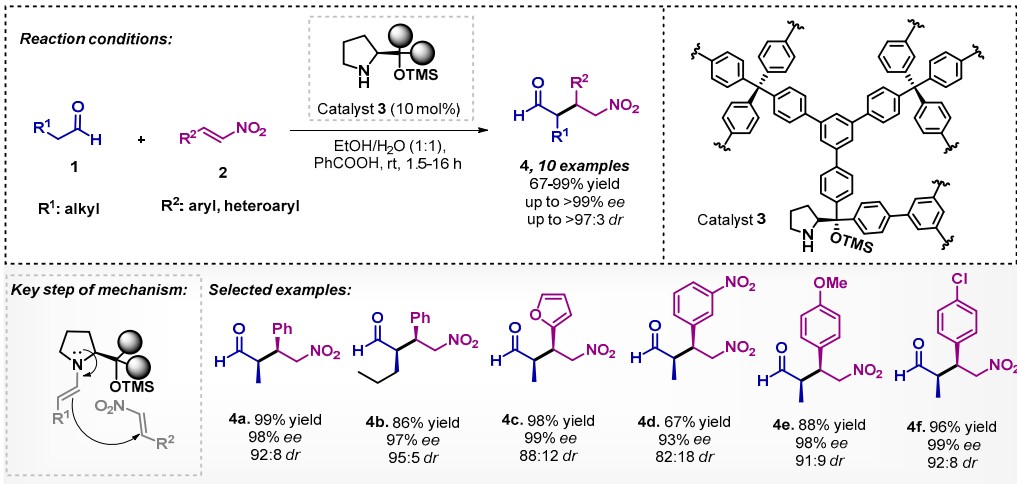

**Scheme 1.** Asymmetric Michael addition using a proline derivative **3** as heterogeneous organocatalyst.

In the same year, Yang and colleagues [34] described the asymmetric aldol reaction using another proline derivative as a heterogeneous catalyst. They developed the chiral catalyst **7** based on silica magnetic microspheres that supported proline, showing high catalytic activity as well as enantio- and diastereoselectivity, resulting in seven different examples. According to the authors, the aldol reaction is probably also catalyzed by the weakly acidic surface hydroxyl groups on the silica shell. Further advantages of this work are the use of ethanol as solvent and reuse of catalyst up to five runs without loss in activity (Scheme 2). It is worthy to note that when chloroform, a non-green solvent, was evaluated in the optimization study, the product **8a** was obtained in only 57% yield and 58% *ee*.

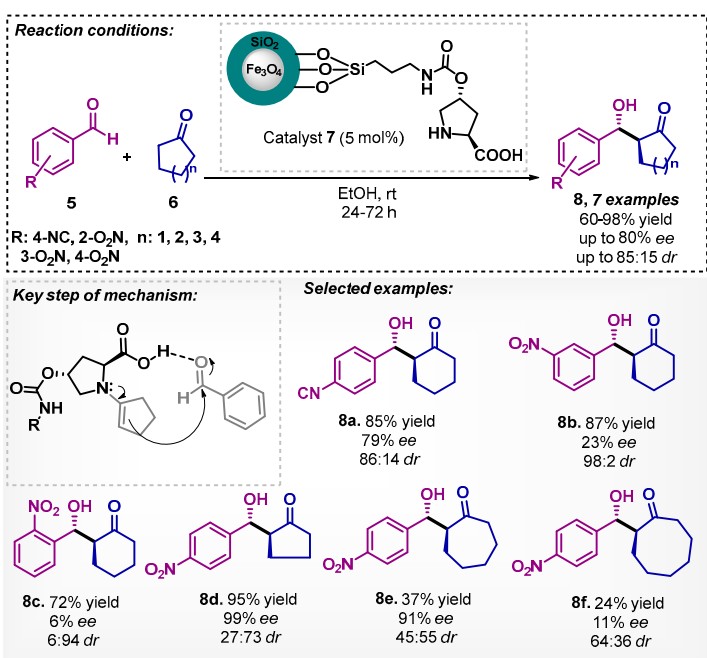

**Scheme 2.** Asymmetric aldol reaction with silica magnetic microspheres supported proline derivative **7**.

The synthesis of pyrans or thiopyrans **12** via multicomponent organocatalyzed reaction was evaluated by Elnagdi and coworkers [35]. In this study, the reaction was carried out with benzaldehyde (**5**), malononitrile (**9**), and compounds possessing activated methylene **10**, using *L*-proline (**11**) as the organocatalyst. Under the optimized conditions, it was possible to obtain eight examples of compound **12**, with yields varying between 60–92% (Scheme 3).

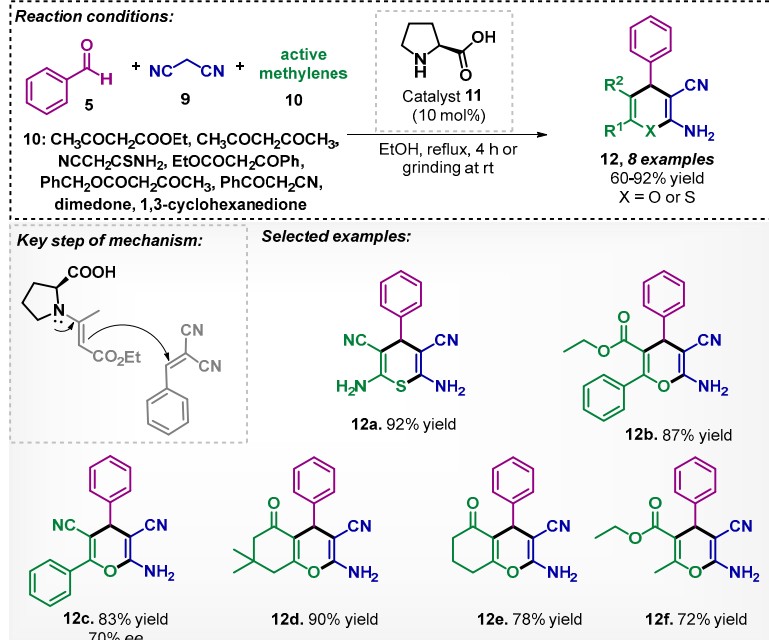

**Scheme 3.** Enantioselective multicomponent reaction for synthesis of pyrans/thiopyrans **12**.

Another use of proline (**11**) as organocatalyst for asymmetric synthesis was described by Dong and colleagues [36]. They combined a one-pot photo- and organocatalysis consisting of an oxidative dearomatization of 2-arylindoles **13**, followed by asymmetric Mannich reaction in ethanol. Due to significant solvent effects on the reactivity of organic reactions,

the authors also tested acetonitrile, methanol, dichloroethane, THF, DMF, and DMSO, but the best yield was obtained with ethanol. The authors explored *L*-proline (Scheme 4) and *D*-proline as organocatalysts to obtain a series of C2-quarternary indolin-3-one derivatives **14** with excellent enantio- and diastereoselectivity.

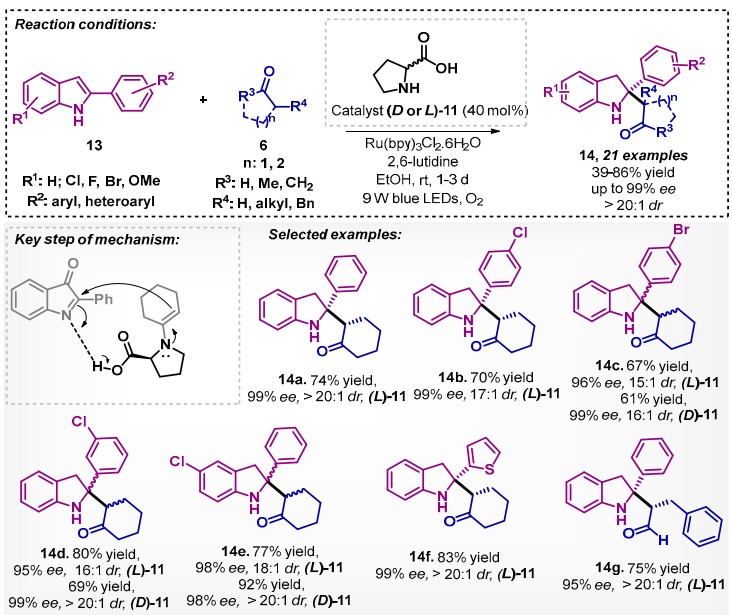

**Scheme 4.** Asymmetric Mannich reaction catalyzed by (*L*)- or (*D*)-proline (**11**).

Moreover, Yadav and Singh [37] used a proline derivative **17** as organocatalyst in the presence of ethanol, exploring a series of prolinamides for the direct asymmetric aldol reaction between isatins **15** and acetone (**16**) to obtain the corresponding derivatives **18**. During the optimization study, other polar solvents such as water and DMF also showed excellent yields (99%) but low to moderate *ee* (38–51%), whereas DCM furnished 96% yield and only 45% *ee*. The authors suggest that the hydrogen bonding between the catalyst and isatin is more efficient, improving the *ee* due to the presence of bromine in the phenyl ring of prolinamide **17** (Scheme 5).

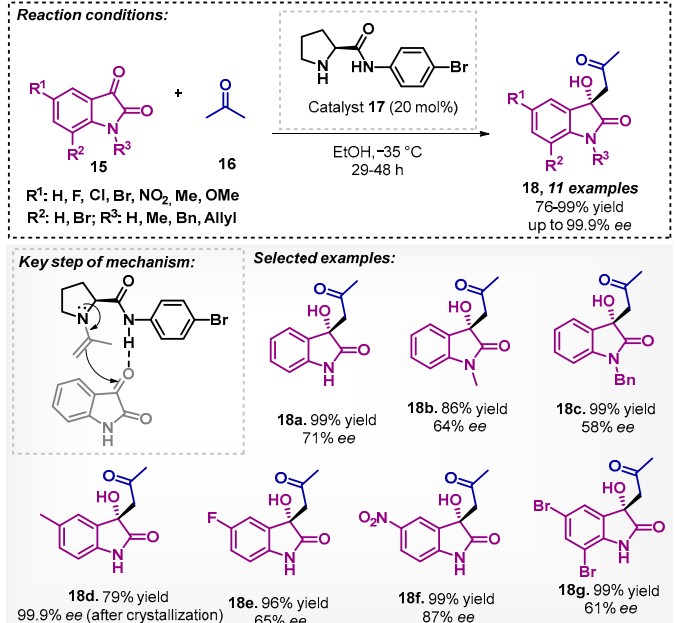

**Scheme 5.** Direct asymmetric aldol reaction catalyzed by prolinamide **17**.

Due to a large number of pharmaceuticals and natural products that contain the $\gamma$-lactam core, Joie and colleagues [38] developed a process to obtain 1,3,5-triarylpyrrolidin-2-ones **23** with three contiguous stereogenic centers. The authors described a one-pot method comprising an aza-Michael/aldol domino reaction of ketoamides **19** and aldehydes **20** in the presence of the Jørgensen–Hayashi catalyst **21** and ethanol as solvent, followed by the addition of stabilized Wittig reagent **22** after 3–16 h, without prior purification, to promote aldehyde olefination. It is worthy to note that chloroform, dichloromethane, and toluene were also tested as solvents, but the key intermediate was not observed. In total, 14 examples of compounds **23** were obtained as a single diastereoisomer with good to excellent enantioselectivities (Scheme 6).

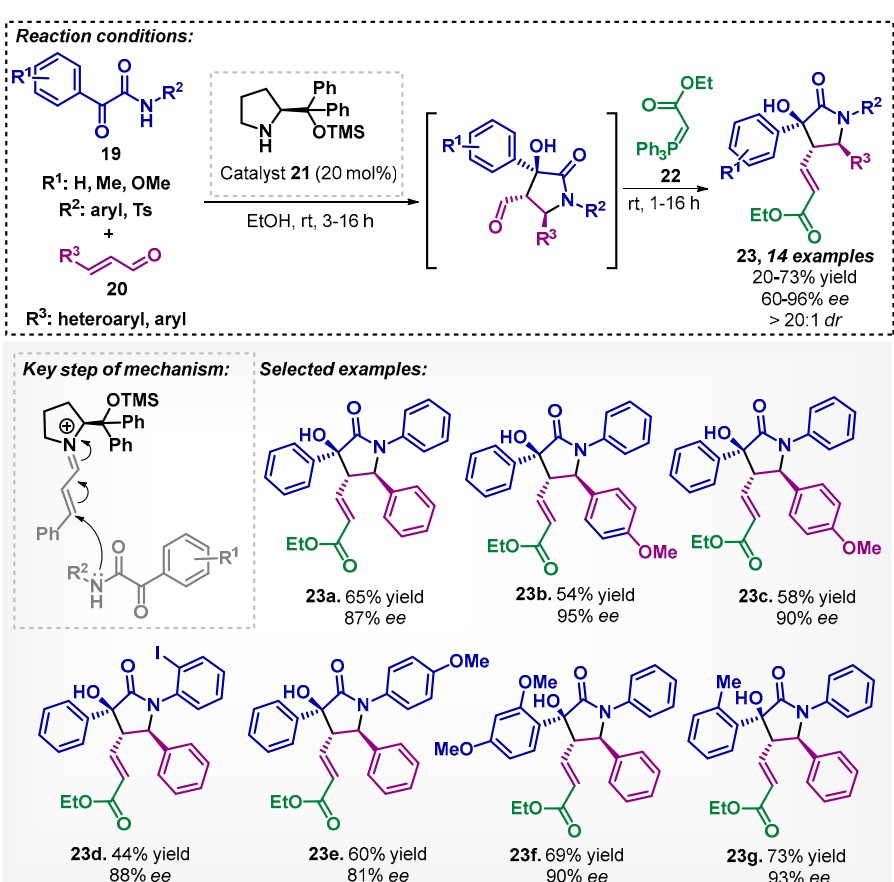

**Scheme 6.** Aza-Michael/Aldol reaction catalyzed by proline derivative **21**.

Later in 2019, Ding and colleagues [39] studied the use of catalyst **25** in asymmetric arylmethylation/$N$-hemicetalization reactions in mild conditions to obtain dihydropyrido [1,2-$\alpha$]indoles **26** with excellent enantioselectivities. The solvent screening included dichloromethane, acetone, toluene, methanol, isopropanol, and ethyl acetate and showed that the Michael addition could be accelerated by protic solvents, and ethanol presented the best performance. In the developed one-pot two steps method, a Michael addition initially occurs in ethanol and DABCO as a base to deprotonate the indole **24**, followed by treatment with TFA (trifluoroacetic acid) (Scheme 7).

Deobald and colleagues [40] developed an asymmetric organocatalytic epoxidation of $\alpha,\beta$-unsaturated aldehyde **20** in a mixture of ethanol/water (3:1) as solvent using a new diarylprolinol silyl ether catalyst **28** with good to excellent enantio- and diastereoselectivities (Scheme 8a). Moreover, the chiral epoxides **29** were employed in a one-pot Passerini reaction without isolation of the epoxides (Scheme 8b), furnishing epoxy-$\alpha$-acyloxycarboxamides **32** that were evaluated as cathepsins inhibitors [41].

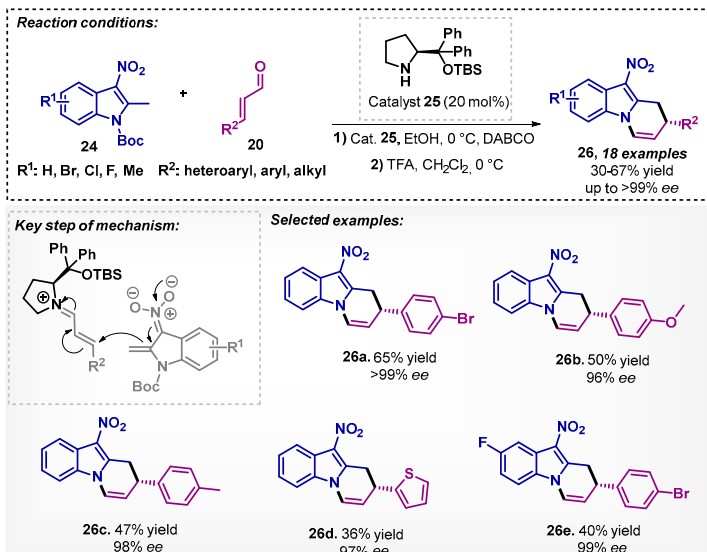

**Scheme 7.** Asymmetric arylmethylation catalyzed by proline derivative **25**.

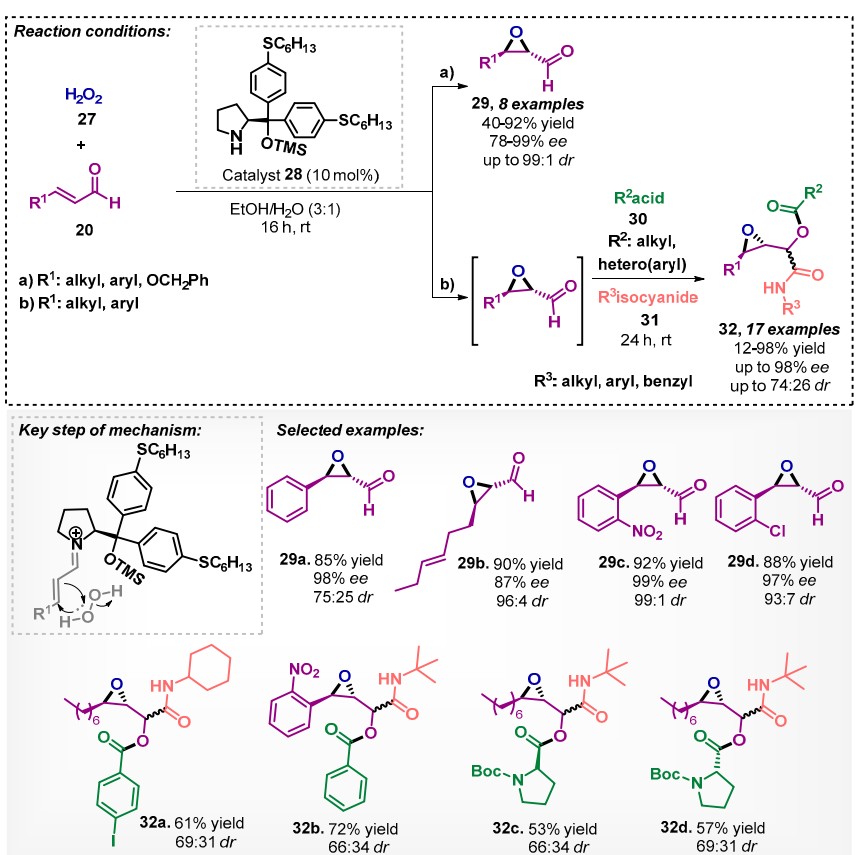

**Scheme 8.** Asymmetric epoxidation catalyzed by diarylprolinol silyl ether **28**.

Santos and colleagues [42] reported the one-pot synthesis of aziridines-α-acyloxycarboxamides **34**. In the studied protocol, the organocatalyst **28** was used in an ethanol/water mixture as solvent to afford the aziridines, which, without isolation, were then submitted to the Passerini reaction (Scheme 9). Under the optimized condition, it was possible to reach a broad scope of 18 aziridine-α-acyloxycarboxamides **34**, with moderate to good yields and up to 94% *ee*. Furthermore, the chiral aziridines were also employed in the Ugi reaction, furnishing new peptidomimetics that were evaluated against cathepsin K [43].

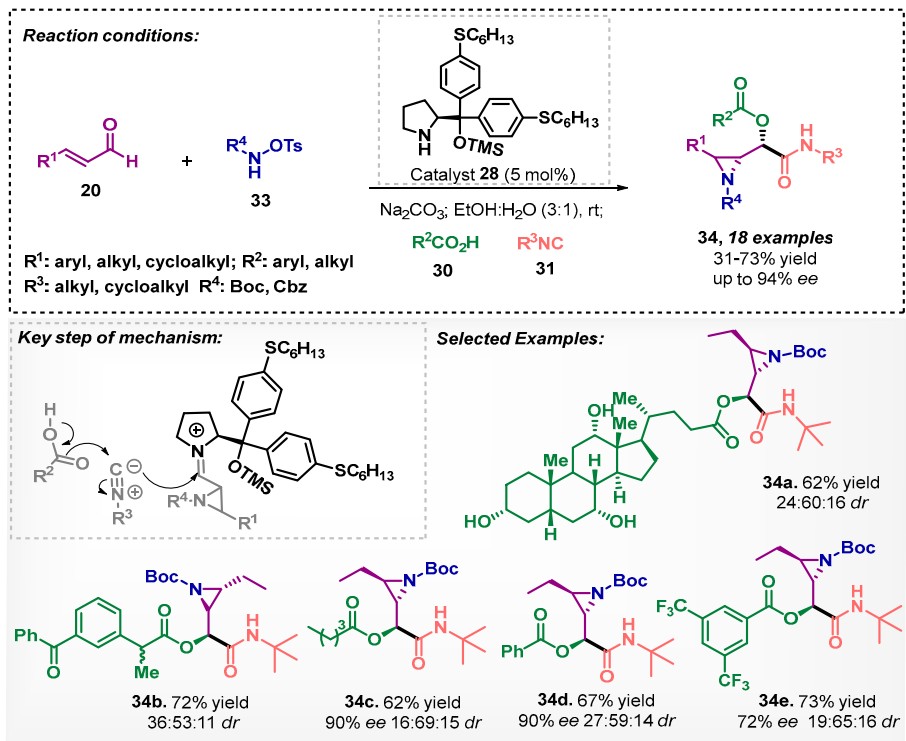

**Scheme 9.** One-pot synthesis of aziridines-*α*-acyloxycarboxamides **34**.

Feu and colleagues [44] investigated an asymmetric Michael addition mediated by organocatalyst **28** in an ethanol/brine mixture as solvent. The method employed *α,β*-unsaturated aldehydes **20** and malonates **35** at room temperature for up to 34 h. In this context, it was possible to obtain nine examples of compounds **36** with moderate to good yields and excellent *ee*. The authors report that after activation of the aldehyde, the iminium ion could be stabilized by the chloride anion (Scheme 10). Moreover, the Michael adducts **36** were used for the synthesis of chiral indoles.

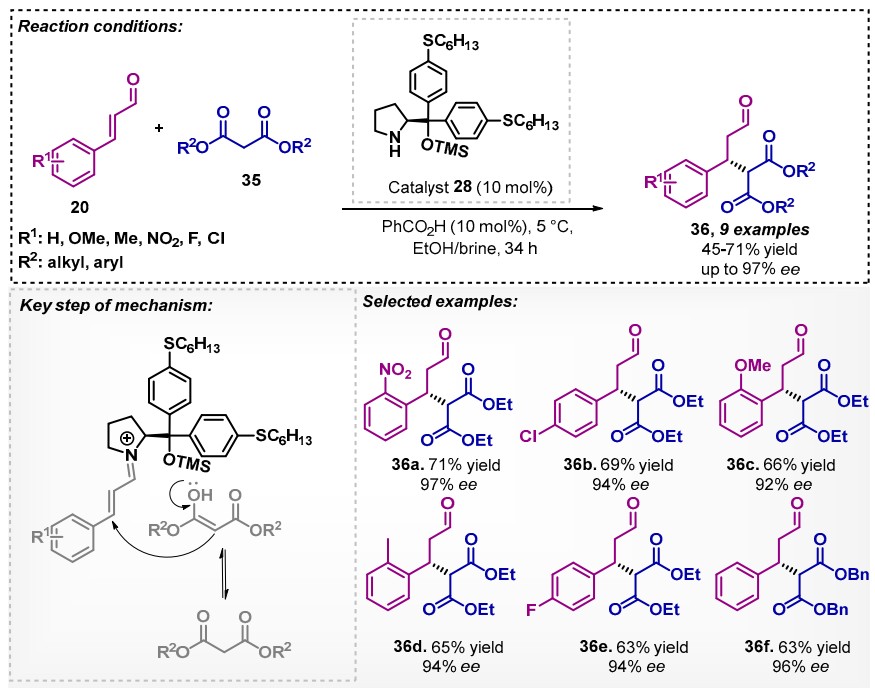

**Scheme 10.** Asymmetric Michael addition using organocatalyst **28**.

Odoh and colleagues [45] studied the asymmetric synthesis of pentasubstituted cyclohexenes **38** from aldehydes **20** and *α*-acetyl-*β*-substituted-*α,β*-unsaturated esters **37**. The optimized condition includes the Jørgensen–Hayashi organocatalyst **21** and Na$_2$CO$_3$ as an additive (Scheme 11). The developed method provided 12 examples of pentasubstituted cyclohexenes with yields between 55–84% in 9–89 h, and excellent diastereo- and enantioselectivities. The authors proposed a transition state to obtain **38**, involving the generation of the iminium ion formed by the reaction between the organocatalyst and the aldehyde **20**, in which it subsequently reacts with the ester **37** to obtain the enamine. Moreover, other solvents were evaluated, for example, CH$_2$Cl$_2$, THF, and toluene, obtaining yields below 10%, and *dr* and *ee* were not determined.

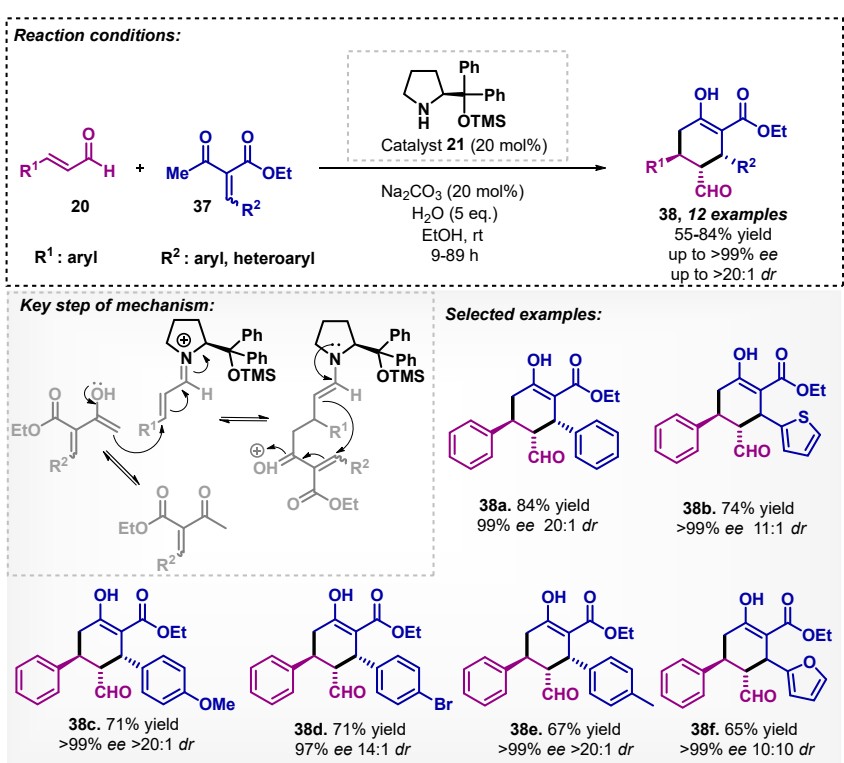

**Scheme 11.** Asymmetric synthesis of pentasubstituted cyclohexenes **38**.

Gellman and colleagues [46] described the synthesis of the peptide (1*R*,2*R*)-2-(aminomethyl)cyclopentanecarboxylic acid **43** in 17 steps (Scheme 12). The key intermediate **41** was efficiently obtained through an asymmetric Michael addition of nitromethane **39** to cycloalkene-1-carboxaldehyde **40** in the presence of ethanol, benzoic acid, and 2,4,6-collidine. This method was based on the previous reports by Hayashi [32] and Wang [33], in which the catalyst **21** and the *α,β*-unsaturated aldehyde originate an iminium ion, resulting in the product **41**, with excellent yield and *ee*, by increasing the catalyst loading and using 2,4,6-collidine for the activation of nitromethane. The *γ*-amino acid **42** could then be obtained from reduction of compound **41**, followed by protection and oxidation.

Using a similar approach, in 2015, Bernardi and colleagues [47] studied the preparation of intermediates that would give access to the core of telaprevir, a serine protease inhibitor of the hepatitis C virus. The authors performed an optimization of the protocol described by Gellman [46] with catalyst *(S)*-**21** and the same substrates. Then, they applied the new optimal condition in the asymmetric addition of amidomalonates **44** to cycloalkene-1-carboxaldehyde **40** with lithium acetate and acetic acid in the presence of ethanol at room temperature. The described method furnished compound *ent*-**45a**, whereas compounds **45a** and **45b** could be obtained with catalyst *(R)*-**21** in moderate to good yields (Scheme 13).

**Scheme 12.** Asymmetric addition of nitromethane to cycloalkene-1-carboxaldehyde with proline derivative catalyst **21**.

**Scheme 13.** Asymmetric addition of amidomalonates **44** to cycloalkene-1-carboxaldehyde promoted by catalyst **21**.

Catalyst **21** was also employed with ethanol as a solvent by Duan and colleagues [48] in the asymmetric formal [2+2] cycloaddition of 2-vinyl pyrroles **46** and cinnamaldehyde **20**, followed by reduction. Based on tandem iminium–enamine activation, the protocol made possible the synthesis of 19 examples with moderate to good yields and enantiomeric excesses (Scheme 14). The authors also tested THF as a solvent, resulting in 40% yield and 90% *ee* for **47a**. Previous reports in the literature showed that [2+2] cycloadditions can be efficiently organocatalyzed in dichloromethane [49] and THF:1,4-dioxane (1:2) mixture [50].

Recently, Zu and coworkers [51] reported the use of cinnamaldehyde **20** and methyl coumalate **48** in a cross-vinylogous Rauhut–Currier reaction with TFA and the proline derivative **25** as catalyst (Scheme 15). The method showed to be efficient in leading to a wide variety of products **49** with good to excellent yields and enantioselectivities. It is noteworthy that when changing the solvent from dichloromethane to ethanol, the reaction time decreased from 72 to 24 h, the stereoselectivity of the reaction was maintained, and the yield increased. In previous studies reported in 2012, the best reaction conditions for the organocatalyzed Rauhut–Currier reaction was in DCM for 24 h [52]. The mechanistic study demonstrated that up to a certain point, two catalyst molecules are involved in the reaction, being one on the formation of the iminium ion intermediate and the other activating nucleophilically methyl coumalate.

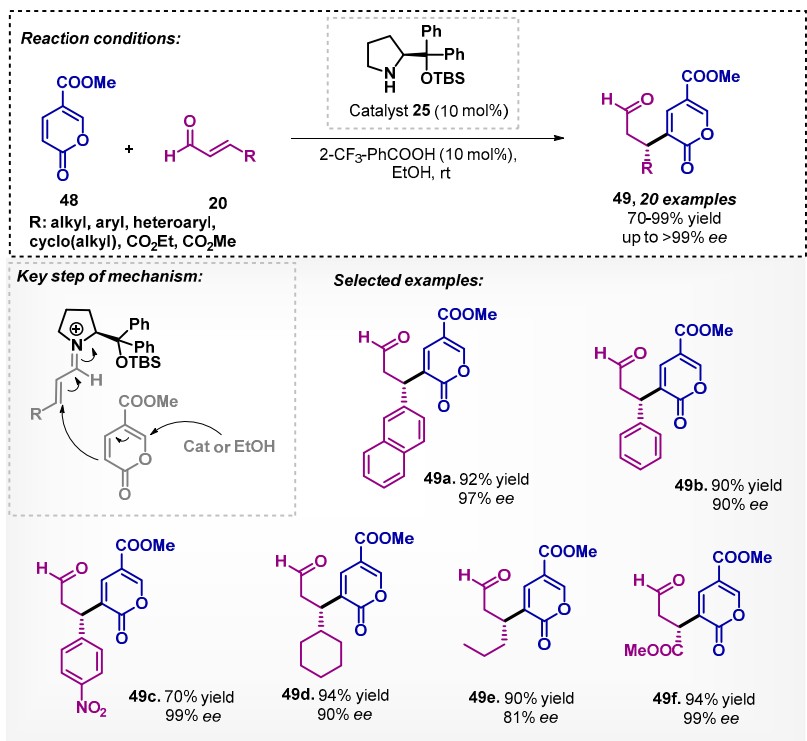

**Scheme 14.** Asymmetric formal [2+2] cycloaddition of 2-vinyl pyrroles **46** and cinnamaldehydes **20** catalyzed by proline-derivative **21**.

**Scheme 15.** Rauhut–Currier reaction of cinnamaldehyde **20** and methyl coumalate **48** catalyzed by proline derivative **25**.

Interestingly, Greco and colleagues [53] developed a protocol considering environmental and economic issues when applying a heterogeneous catalyst **50** and a EtOH:H$_2$O mixture as the solvent in a continuous flow regime for a stereoselective aldol reaction of cyclic ketones **6** with aromatic aldehydes **5**. Immobilization of the organocatalyst was

successfully developed in a polystyrene monolithic column with the (S)-5-(pyrrolidin-2-yl)-1H-tetrazoles. The adducts were obtained through a complete conversion of aldehydes **5** with good to excellent enantio- and diastereoisomeric controls. The immobilized catalyst **50** maintained its stereoselective performance in flow for 120 h (Scheme 16). Although, in this protocol, only water was fundamental for the enantioselectivity of the reaction with *ee* of 95%, against 84% *ee* in ethanol, the EtOH:$H_2O$ mixture was necessary for better substrates solubilization. Organocatalyzed aldol reactions have had widespread success in building interesting molecules, yet toxic solvents such as DCM [54,55], chloroform [56], and DMF [57] are still continually used.

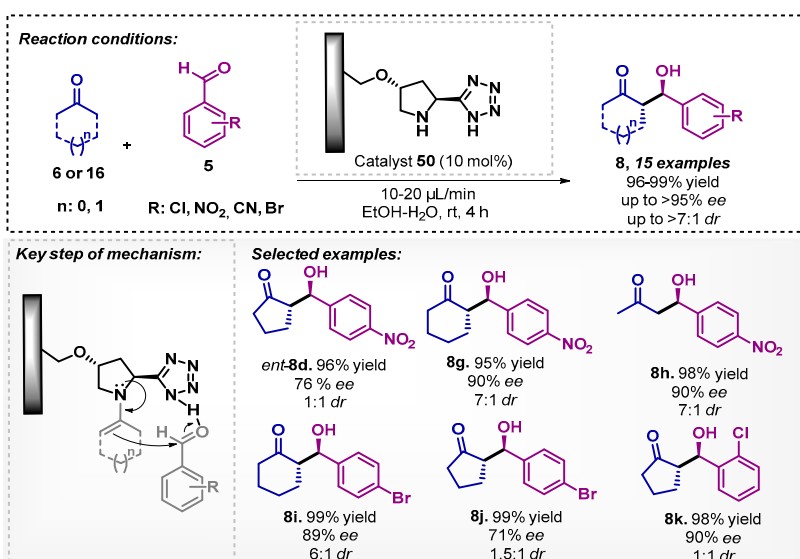

**Scheme 16.** Asymmetric aldol reaction of cyclic ketones **6** and aromatic aldehydes **5** with heterogeneous catalyst **50**.

Concerning the primary amine catalysts, they also proved to be efficient in the presence of ethanol. For example, in 2013, Wang and colleagues [58] investigated the use of diphenyl-1,2-ethanediamine as a chiral organocatalyst in the enantioselective Michael addition of aryl methyl ketones **51** with 2-furanones **52** and *p*-toluenesulfonic acid as co-catalyst. The presence of two primary amines activated both substrates, forming an enamine with **51** and a hydrogen bonding with **52**. The β-lactones could be obtained with yields between 63–92% and excellent enantio- and diastereoselectivities (Scheme 17).

### 2.1.2. Activation Via Hydrogen Bonding

The usual organocatalysts for activation via hydrogen bonding are thioureas, squaramides and phosphoric acids [59]. The energy of the substrate LUMO decreases, resulting in a more electrophilic species that is susceptible to nucleophilic attacks. The hydrogen bonding also causes an additional stabilization of the transition state [1,3].

The asymmetric Michael addition of ketones **55** to β-nitrostyrenes **2** was investigated by Pramanik's group [60] to obtain the corresponding adducts **57**. In the developed method, proline-oxadiazolone organocatalyst **56** was synthesized and applied in the presence of ethanol as a solvent and triethylamine as a base, allowing for obtaining 18 examples of Michael adducts **57**, reaching yields of up to 97% with excellent enantio- and diastereoselectivities. It is worthy to note that other solvents, such as DMSO, THF, DCE, $CHCl_3$, and 2-propanol, were also tested; however, they did not present better results when compared to ethanol. The authors proposed that the selectivity was due to an electrostatic interaction between the nitro group and nitrogen of pyrrolidine ring, including an extended hydrogen bonding in the transition state (Scheme 18).

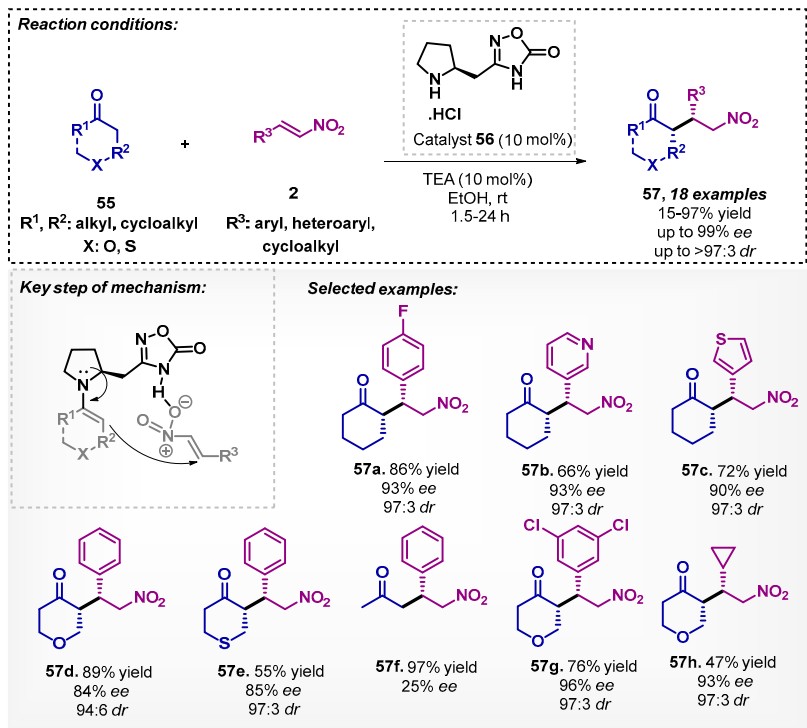

**Scheme 17.** Enantioselective Michael addition of aryl methyl ketones **51** with 2-furanones **52** catalyzed by diphenyl-1,2-ethanediamine **53**.

**Scheme 18.** Pyrrolidine-oxadiazolone organocatalyst **56** in asymmetric Michael reaction.

Bhusare and co-workers [61] developed a convenient protocol for the synthesis of Baylis–Hillman adducts using a proline derivative **59** as organocatalyst. In the screening of solvents, water and dioxane also were evaluated, but ethanol provided a reduction in reaction time (from 48 to 12 h). The *α*-methylene-*β*-hydroxy derivatives **60** were obtained

with good to high yields and *ee* up to 96% (Scheme 19). The authors described that insertion of acetyl group in the active nitrogen of the catalyst restricts the pathway for formation of imine intermediate in the transition state, thus, allowing the hydrogen bonding to promote the reaction.

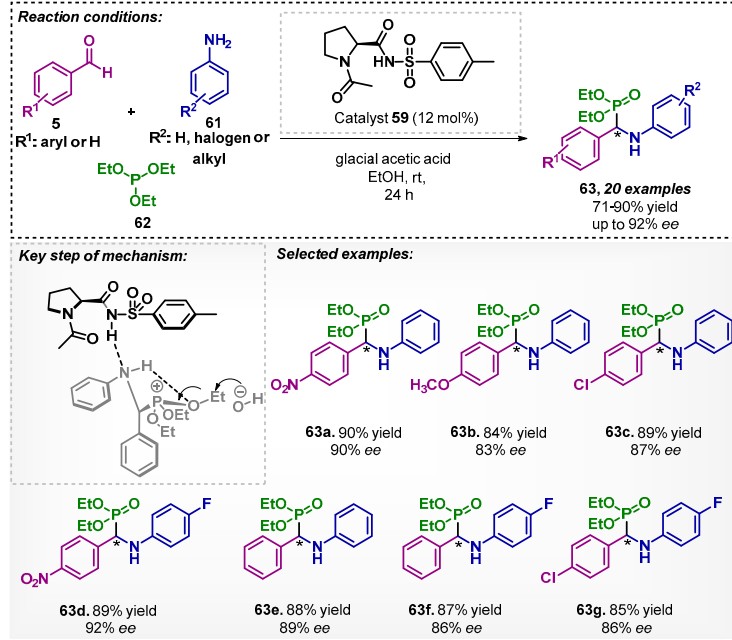

**Scheme 19.** Asymmetric synthesis of α-methylene-β-hydroxy derivatives with organocatalyst **59**.

The same group also studied the asymmetric synthesis of α-aminophosphonates **63** via a one-pot reaction (Scheme 20) [62]. The transformation involves aldehydes **5**, anilines **61,** and triethyl phosphite (**62**) as starting materials and was catalyzed by organocatalyst **59** in presence of glacial acetic acid. The method allowed for obtaining 20 derivatives of α-aminophosphonates **63** with good to excellent yields and *ee*. The authors proposed a mechanism pathway in which the formation of the imine initially occurs, then, due to the hydrogen bonding with the organocatalyst in the transition state, compounds **63** are formed. In this work, water and dioxane were also evaluated as solvents, however, showing lower yields and *ee*.

**Scheme 20.** Enantioselective synthesis of α-aminophosphonates **63**.

Bhusare's group [62] investigated the synthesis of β-malonophosphonates **65** derivatives using trialkyl phosphites (**62**) and α,β-unsaturated malononitriles **64**, mediated by proline derivative **59** as organocatalyst. In this perspective, the method allowed for obtaining 13 examples of β-malonophosphonate derivatives **65** with yields of up to 85% and *ee* reaching up to 78% (Scheme 21). In this protocol, other solvents were studied, such as water and toluene, and it was found that, among them, the most effective for the synthesis of β-malonophosphonate derivatives was DCM, furnishing the product of interest **65a** with only 40% yield and 34% *ee*.

**Scheme 21.** Organocatalyst **59** promoted synthesis of chiral β-malonophosphonates **65**.

Bhusare and colleagues [63] also developed a method for the asymmetric Baylis–Hillman reaction. The protocol was guided by organocatalyst **66**, and aromatic aldehydes **5** and methyl acrylate (**58**) were adopted as starting materials to obtain the β-hydroxy acrylates **60**. The reaction was carried out at room temperature, in up to 19 h, furnishing 10 examples of the β-hydroxy acrylates **60** with yields varying between 75–90% and excellent enantioselectivity (reaching up to 100% *ee*) (Scheme 22). Other solvents were investigated, such as water, toluene and DMF, and among them, DCM was the most efficient furnishing product *ent*-**60b** with only 62% yield and 59% *ee*. The proposed mechanism pathway involves a combination of hydrogen bonding of the organocatalyst with the enol form of methyl acrylate (**58**) and a covalent activation of the aldehyde **5** by the proline moiety, resulting in the enamine.

Smith's group [64] developed a protocol for the enantioselective synthesis of substituted dihydropyran derivatives **70**, in which ethanol was used as a solvent to promote the reaction between 4,4,4-trifluorobutenones (**68**) and ethyl 2,3-butadienoate (**67**) via [4+2] cycloaddition. In this method, using quinidine (**69**) as organocatalyst at room temperature and for up to two days, it was possible to obtain 12 derivatives of dihydropyrans **70**, with yields varying between 34–71% and reaching *ee* of up to 90% (Scheme 23). Other solvents were evaluated, and the most efficient was toluene, furnishing the desired product in 85% yield and 79% *ee*. In the scope and limitations, different trifluoromethylenone derivatives were also explored using toluene or acetone as solvents.

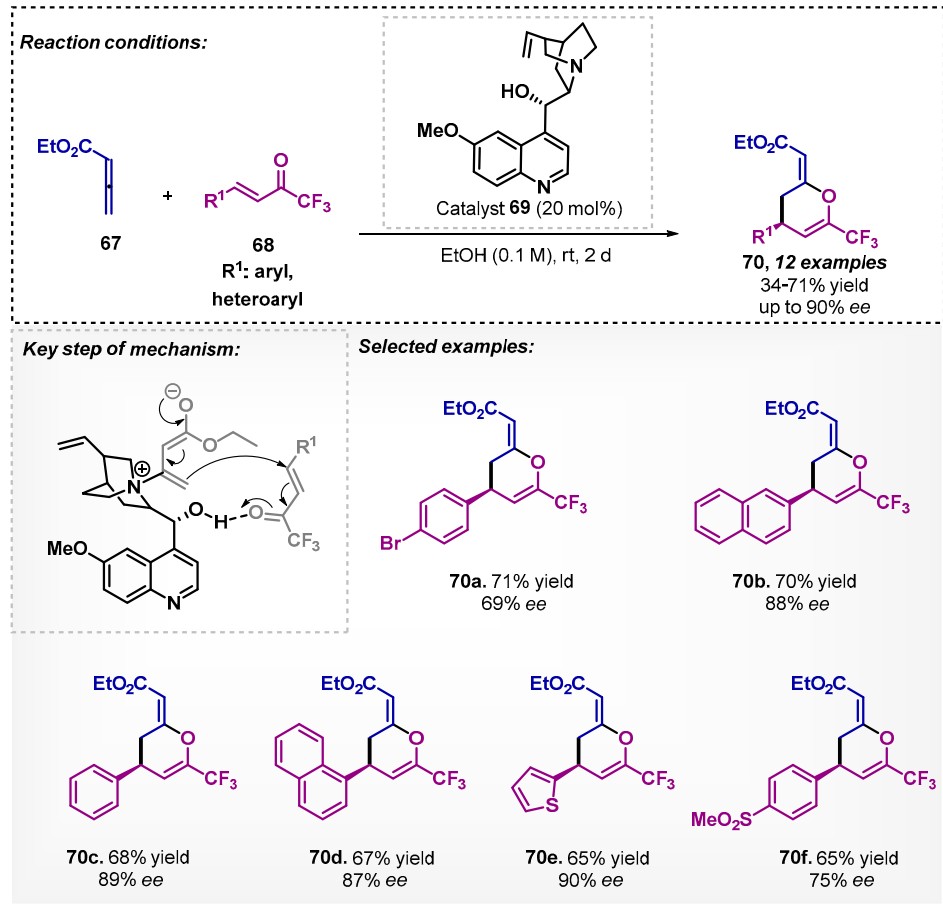

**Scheme 22.** Organocatalyzed asymmetric Baylis–Hillman reaction.

**Scheme 23.** Enantioselective protocol for the synthesis of dihydropyran derivatives **70**.

Zlotin's group [65] described the Michael reaction between kojic acid derivatives **71** and nitroolefins **2** using only 1 mol% of the $C_2$-symmetric tertiary amine-squaramide **72** as organocatalyst in ethanol or water as solvents, affording adducts **73** in high yields and *ee* (Scheme 24). Other solvents, such as DCM, THF, MeOH, and EtOAc, were also evaluated,

but the best results were obtained with ethanol. Interestingly, it was also possible to recover and reuse the catalyst in up to 7 cycles.

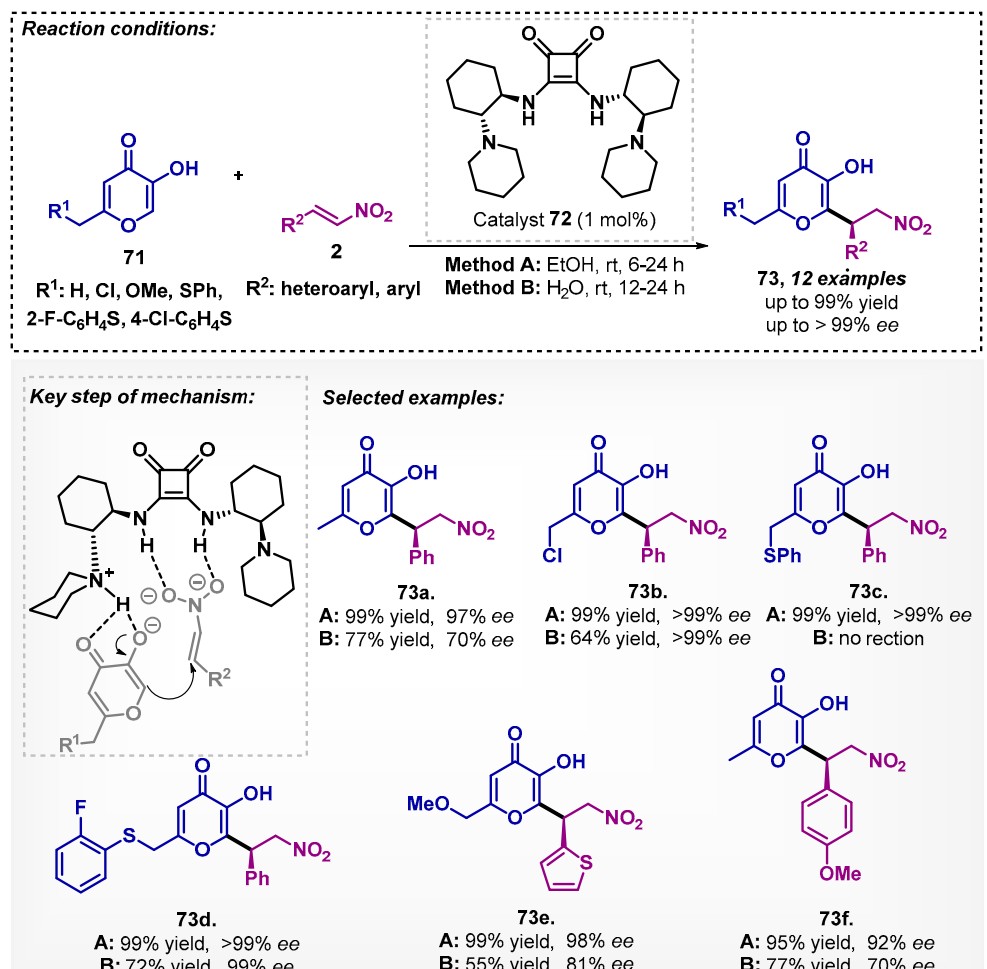

**Scheme 24.** Asymmetric Michael reaction catalyzed by recyclable squaramide **72**.

Later, in 2021, the same group [66] demonstrated that the squaramide **72** could be applied efficiently with 3-hydroxychromen-4-ones **74** using ethanol as solvent, since the stereoinduction decreased in aprotic solvents such as EtOAc, PhMe, and hexane. The corresponding Michael adducts **75** were obtained with high yields and enantioselectivities, and the organocatalyst could be reused in 10 cycles without significant loss of activity (Scheme 25). Differently from the mechanism shown in Scheme 24, the authors proposed that in addition to the deprotonation of chromone **74** by tertiary amine group of the catalyst, a strong hydrogen bonding with the electron-deficient olefin **2** also occurs, probably due to the steric hindrance of the substituent $R^3$ with the squaramide substituents.

Aral and colleagues [67] have synthesized a range of chiral β-aminoalcohols that were evaluated as organocatalysts for opening of glycidol (**77**) with phenols **76**. In this perspective, using catalyst **78** in ethanol as a solvent, the method afforded three examples of 3-aryloxy-1,2-propanediols **79**, with yields of 45–65% and *ee* of up to 58%. The authors proposed a mechanism pathway based on the activation of the glycidol ring followed by phenol attack, which was promoted by hydrogen bonding with the catalyst (Scheme 26).

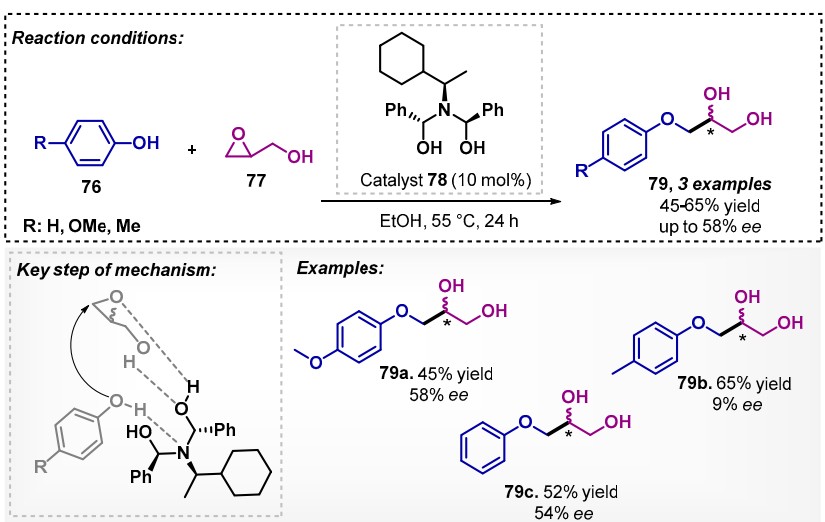

**Scheme 25.** Asymmetric conjugate addition catalyzed by recyclable squaramide **72**.

**Scheme 26.** Synthesis of 3-aryloxy-1,2-propanediol derivatives **79** mediated by organocatalyst **78**.

### 2.1.3. Activation Via Ion Pair

Non-covalent organocatalysis can also occur through an ion pair of the catalyst with the substrate. The electrostatic interaction between the cationic and anionic species generated in situ causes a steric hindrance on one of the faces of the electrophile. For example, it allows a stereoselective approach of the nucleophile. This process may be performed via phase transfer catalysis (PTC), when the cationic organocatalyst interacts with the substrate at the aqueous or solid phase, forming an ion pair intermediate soluble in organic solvent and, thus, enabling chemical transformations [68]. On the other hand, it might occur via an

asymmetric counter anion directed catalysis (ACDC) reaction, proceeding through an ion pairing with a chiral anion provided by the catalyst [69].

Ashokkumar and Siva [70] developed a tetrafunctional cinchonine derivative **80,** resulting in a protocol with low catalyst loading (1 mol%), short reaction time (30 min), good yields, and excellent *ee*. The success of this method in the asymmetric conjugate addition of malonates **9** or **35** to nitroolefins **2** is due to the multiple active sites that are present in the catalyst, which increase the number of ion pair interactions with the substrates (Scheme 27). During the optimization study, the following solvents were evaluated: MeOH, EtOH, acetone, DCM, CHCl₃, THF, and toluene, and the results showed that the alcohols were the best for this Michael addition reaction.

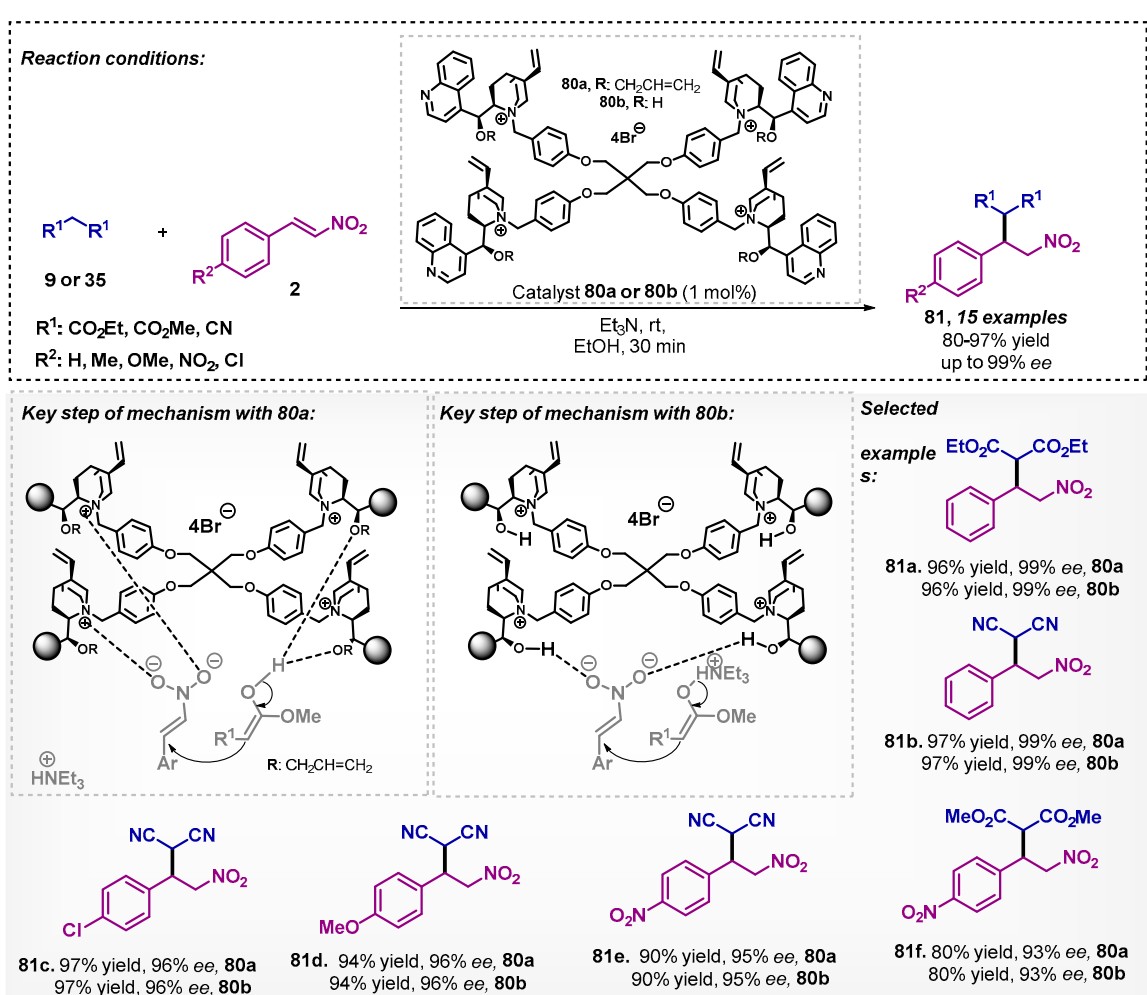

**Scheme 27.** Michael addition reaction of malonates **9** or **35** to nitroolefin **2** with tetrafunctional organocatalysts **80a** and **80b**.

Ashokkumar and colleagues [71] studied a Knoevenagel condensation between different *α*-branched aldehydes **1** with various malonates **35**. The reaction was explored through dynamic kinetic resolution and the organocatalyst was a cinchona derivative **82** at room temperature. The scope of this method was demonstrated by 16 examples with yields between 85–97% and *ee* reaching up to 99%. The authors also explored these adducts to obtain *γ*-alkyl-substituted amides (Scheme 28). In this work, a screening of solvents was carried out, and it was noted that, among the non-biobased evaluated, the best one was DMF, furnishing the desired product in only 74% yield and 69% *ee*.

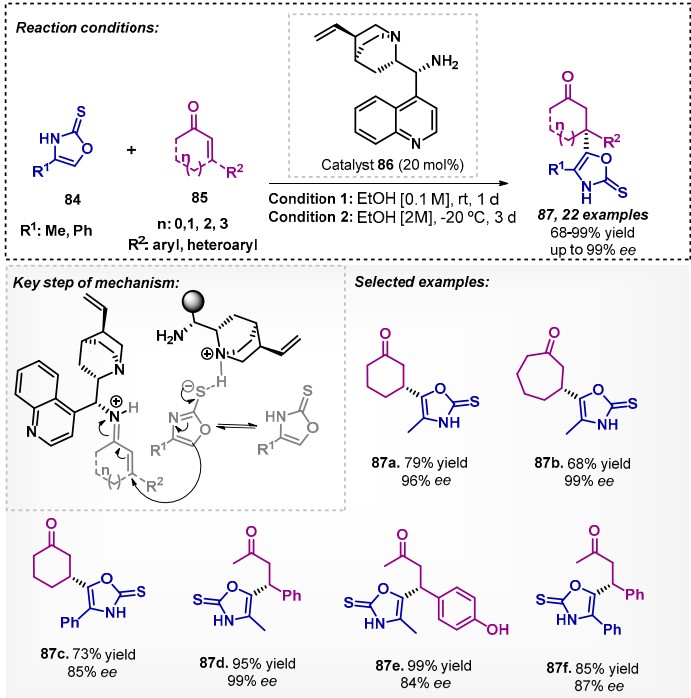

**Scheme 28.** Knoevenagel condensation mediated by a cinchona-derived organocatalyst **82**.

In 2018, Silva and colleagues [72] described a novel protocol for obtaining functional-ized oxazol-2(3*H*)-thiones **87** using 9-amino-9-deoxy-epi-cinchonine (**86**) as a bifunctional organocatalyst (Scheme 29). When the authors employed *i*-PrOH as a solvent, the results encouraged the search for other alcohols, and ethanol was selected to the asymmetric Michael addition without any additive, and compounds **87** could be efficiently obtained with good to excellent yields and *ee*. It is noteworthy that in this Michael addition protocol, DCM and toluene proved to be inefficient, furnishing compound **87a** in only 10 and 15% yields, respectively, although with excellent enantioselectivities. The mechanistic study showed that more than one molecule of the chiral catalyst is likely to be involved in the transition state, generating an ion-pair from an acid-base equilibrium of the catalyst with **84** and the formation of the iminium ion with **85**.

**Scheme 29.** Addition of oxazole-2(3*H*)-thiones **84** to *α*,*β*-unsaturated ketones **85** with 9-amino-9-deoxy-epi-cinchonine.

Liu's group [73] carried out the study of aldol reactions mediated by asymmetric counter anion directed catalysis (ACDC) with *trans*-cyclohexanediamine *L*-tartrate salt (**88**). In that regard, there is a combination of the achiral iminium catalyst and chiral Brønsted acid catalyst. The method was developed with aromatic aldehydes **5** and cyclic ketones **6** at room temperature, allowing for obtaining six examples of compounds **8** with yields ranging from 73–99%, good *ee*, and excellent diastereoselectivities (Scheme 30). The authors evaluated other solvents, such as DCM, DMF, and THF, and surprisingly, none of them led to obtaining the desired derivatives **8**.

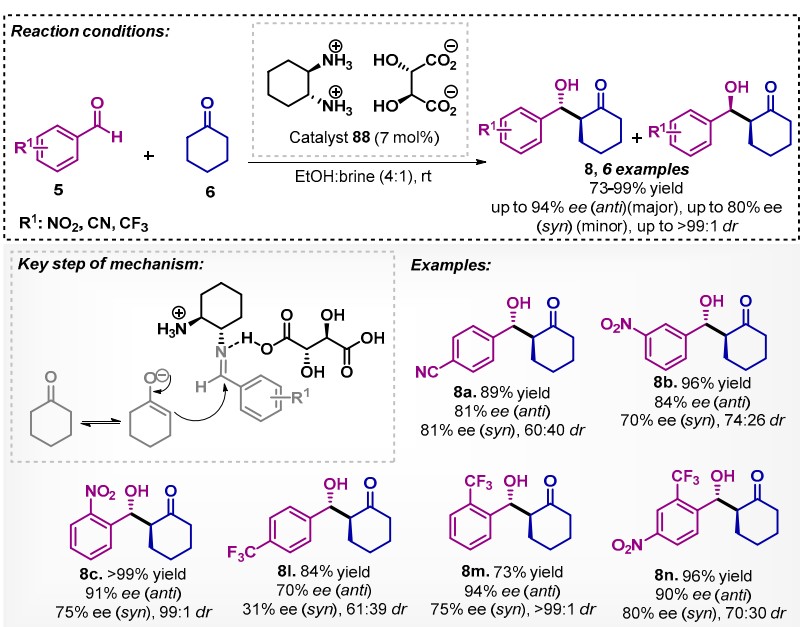

**Scheme 30.** Asymmetrical aldol reaction mediated by catalyst **88**.

## 2.2. Methyltetrahydrofuran

2-Methyltetrahydrofuran (2-MeTHF) is obtained from lignocelluloses, a renewable feedstock, where the predominant source is the dry plants residue. The hemicellulose and cellulose present in the raw material are converted to furfural and levulinic acid, the main precursors of 2-MeTHF [29].

This bio-based solvent, in most cases, reproduces or exceeds the results obtained with THF, thus, being an effective alternative. Although 2-MeTHF demonstrates satisfactory and competitive results and even has lower toxicity, it is observed that, in many works, this solvent is not even applied in solvent screening, which perpetuates the use of THF as the main source of ether-based solvent in organocatalysis. In addition, this commercially available solvent has good stability in acidic and basic environments, low miscibility with water, and biodegradability, which makes it a viable substitute for dichloromethane and toluene [20]. It can still be dried without the need for dangerous drying agents, and due to the clean work-up, allows easy recovery of the product [74,75].

### 2.2.1. Catalysis Via Covalent Bonding

The asymmetric Michael addition reaction of α,α-dicyanoalkenes **64** to 2-enoylpyridine *N*-oxides was first reported in 2014 by Singh's group [76] using toluene and bifunctional thiourea catalyst. In 2019, Martelli and colleagues [77] reported the reaction of chalcones **89** with α,α-dicyanoalkenes **64** with 9-amino-9-deoxy-epi-cinchonine (**90**) as catalyst in the presence of 2-MeTHF with excellent enantio- and diastereoselectivities. The Michael adducts **91** still showed promising results when evaluated against *Plasmodium falciparum* with low cytotoxic activity (Scheme 31).

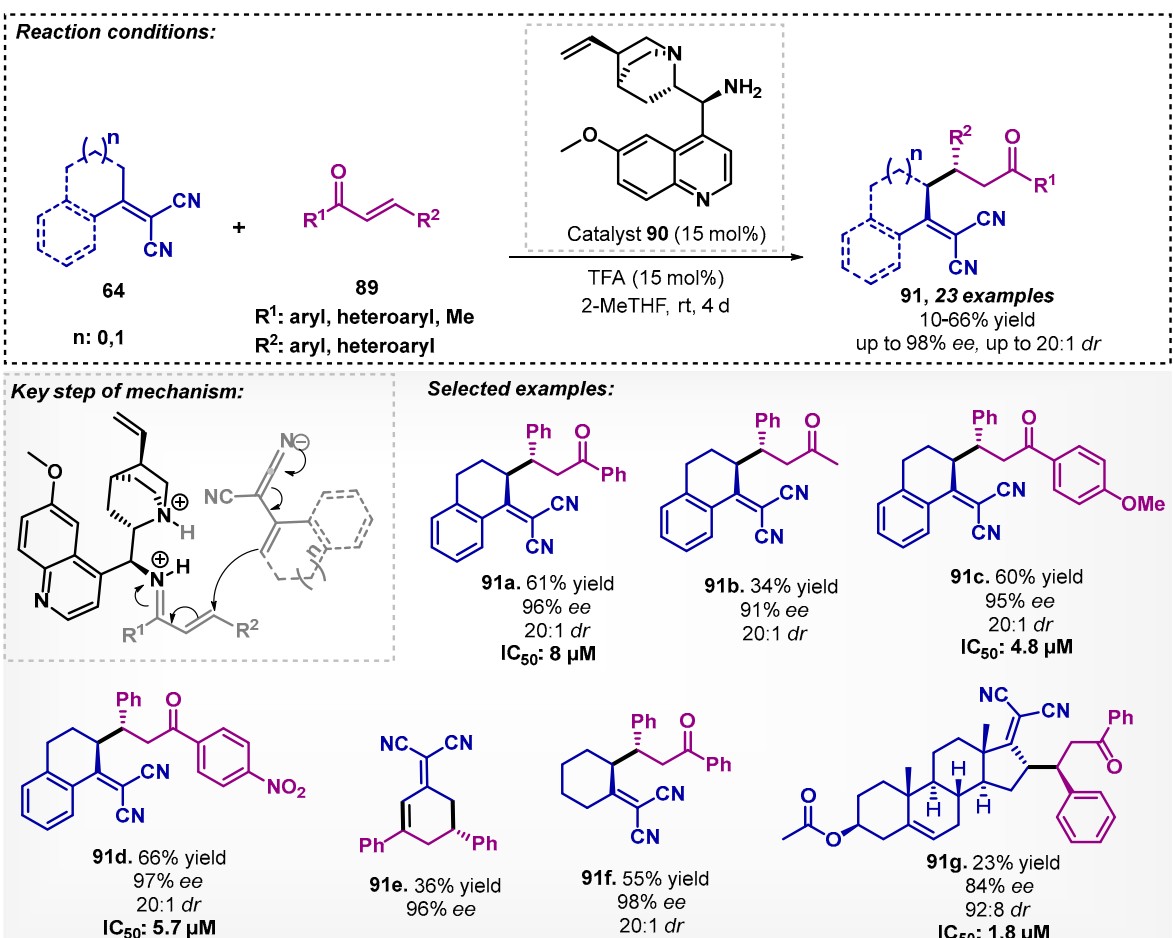

**Scheme 31.** Michael adducts from the reaction of α,α-dicyanoalkenes **64** with chalcones **89** catalyzed by cinchonine **90** and their corresponding *P. falciparum* inhibition.

Cañellas and colleagues [78] developed a protocol for the synthesis of chiral enones **95** via a Robinson annulation reaction using the organocatalyst **94** based on a diamine supported on polystyrene (PS). This method allowed for obtaining 14 examples of bicyclic chiral enones with moderate to excellent yields and *ee* reaching up to 93% in 1–12 h (Scheme 32). The structural peculiarity of substrate **93** allows a simultaneous formation of enamine and imine in the transition state. The authors, during the optimization study, evaluated other solvents, such as DCM and DMF, and among them, THF presented the best result, with 90% yield and 91% *ee* to produce **95c**. The supported catalyst **94**, in batch mode, could be recovered and reused in up to 10 cycles and was also efficiently employed in a 24 h continuous flow experiment.

Huang and colleagues [79] designed a chiral proline derivative **96** as catalyst to apply in the asymmetric aldol reaction between aldehydes **5** and ketones **6** or **16**. They tested the effect of solvents (CH$_2$Cl$_2$, EtOAc, CH$_3$OH, DMSO, THF, α,α,α-trifluorotoluene, brine, 2-methyl-*tert*-butanol and 2-Me-THF) and found high chemical yields and good enantio- and moderate diastereoselectivities when 2-MeTHF or brine were used in the presence of 4-nitrobenzoic acid as an additive (Scheme 33). In some cases, with 2-MeTHF, it was possible to observe the formation of large amounts of the stable imidazolidinone intermediate without the formation of the desired product **8**, but when brine was used, the formation of this intermediate was significantly suppressed.

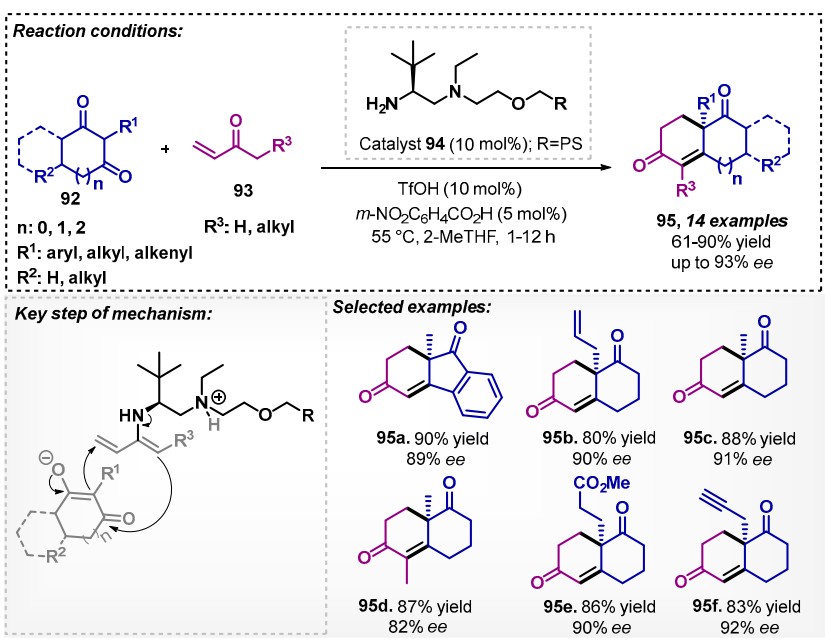

**Scheme 32.** Synthesis of enones **95** via organocatalyzed Robinson annulation.

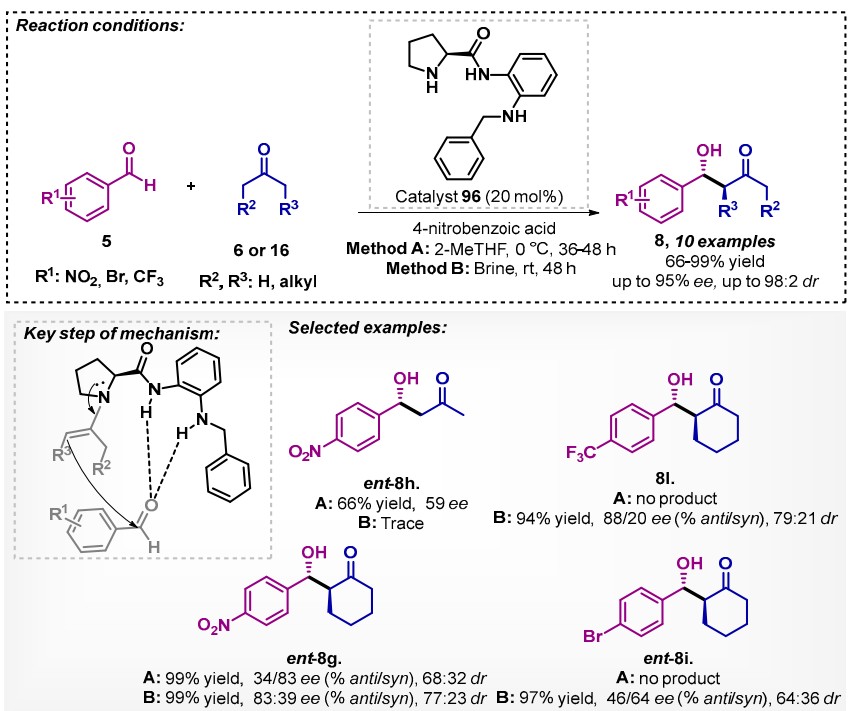

**Scheme 33.** Asymmetric aldol reaction catalyzed by chiral prolinamide **96**.

### 2.2.2. Activation Via Non-Covalent Bonding

The applicability of 2-MeTHF in organocatalysis with activation via non-covalent bonding was described for the enantioselective Michael/hemiketalization addition of hydroxycoumarins **97** to enones **89** by Šebesta and colleagues [80]. It is noteworthy that even though they obtained a yield of 86% and 96% *ee* in dioxane-acetonitrile for **99e**, they considered the impact of this solvent on health and the environment, thus, examining different bio-based solvents, and recognized 2-MeTHF as the best option. The developed protocol resulted in a series of hydroxyketones, including *(R)*-warfarin, that quickly underwent an equilibrium for their hemiketal form **99**, using the squaramide **98** as a catalyst. The

formation of an iminium ion intermediate with enone and hydrogen bonding with hydroxycoumarin led the reaction with low to excellent yields and good to excellent enantiomeric control (Scheme 34).

**Scheme 34.** Michael addition of 4-hydroxycoumarin **97** to enones **89**.

### 2.3. Ethyl Acetate

Ethyl acetate is biodegradable, with relatively low toxicity and low cost, thus, being safe for humans and the environment, resulting in a sustainable alternative to benzene, acetone and dichloromethane [29].

The production of bio-based ethyl acetate is closely linked to ethanol, considering that it is one of the main raw materials, since bio-ethanol and acetic acid in the presence of lipases are efficiently converted to this solvent. However, the major production of ethyl acetate is still from non-renewable sources, natural gas and crude oil, where high energy consumption is required and with the formation of hazardous waste [81].

#### 2.3.1. Catalysis Via Covalent Bonding

Due to the broad spectrum of biological activities assigned to coumarins, Lee and coworkers [82] studied the asymmetric Michael reactions of ketones **6** or **55** to 3-aroylcoumarins **100** to obtain enantiomerically enriched coumarin derivatives **102**. The authors tested a series of chiral catalysts, and according to the results obtained, it is possible to note that differently from primary amines, secondary amines do not show product formation, and this fact could be attributed to the low reactivity between the catalyst and ketone **6** or **55**. In the solvent screening, no improvement of enantioselectivity was observed when they used either polar (EtOAc, THF, and MeOH) or non-polar (DCM and toluene) solvents, thus, the authors proceeded with the optimization using EtOAc. Thus, they developed an efficient method for enantioselective synthesis of chroman-2-ones **102** based on quinidine derivative **101** as catalyst in presence of ethyl acetate as solvent, with yields between 10–98% and high enantio- and diastereoselectivities (Scheme 35). It stands out that *ent*-**102d** could be prepared with 88% *ee* in the same conditions employing catalyst **90** (Scheme 36).

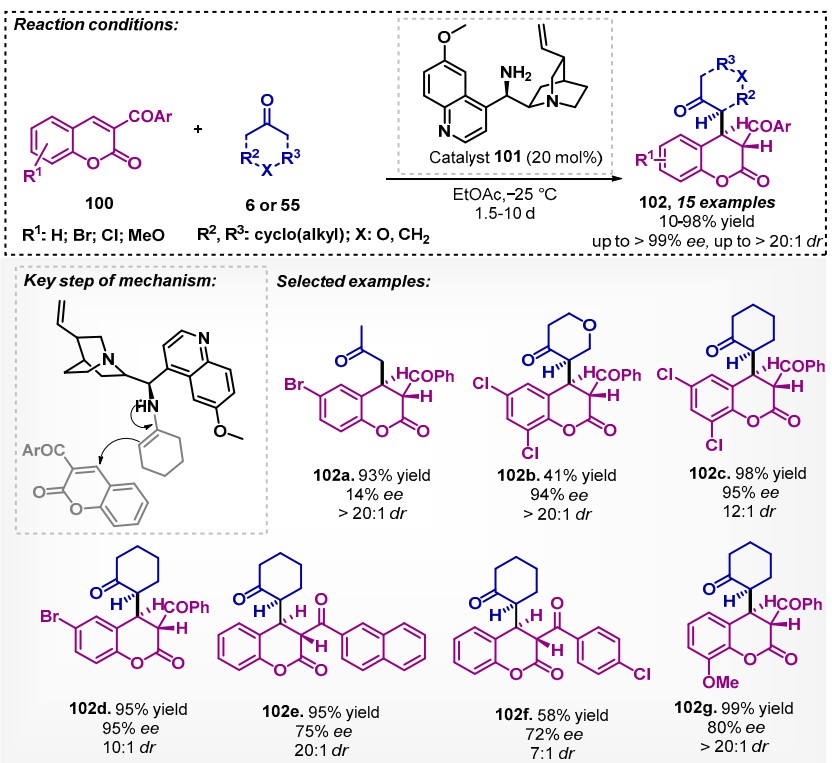

**Scheme 35.** Asymmetric Michael addition catalyzed by primary amine **101**.

**Scheme 36.** Asymmetric Michael addition reaction to obtain *ent*-**102d**.

Recently, based on the importance of the pharmaceutical intermediate (*R*)-pantolactone, Du and colleagues [83] reported the enantioselective aldol reaction of *α*-branched aldehydes **1** and glyoxylate derivatives **103**, using phenolic *tert*-leucinamide **104** as an organocatalyst in EtOAc. In this aldol reaction, EtOAc (91% yield and 93% *ee*) showed better results than the conventional solvent DCM (86% yield and 89% *ee*) for **105a**. According to the proposed transition state, the catalyst promotes the reaction via hydrogen bonding with the enamine intermediate and glyoxylate **103**, furnishing products **105** with up to 97% *ee* and 91% yield. The authors also highlight the efficiency of the method in maintaining the yield and enantioselectivity of *(R)*-pantolactone **105a** at a scale of 50 mmol (Scheme 37).

The asymmetric Michael addition of aldehydes **106** to maleimides **107** was studied by Kozma and colleagues [84] to obtain succinimides **109**. In the developed method, chiral heterogeneous organocatalysts were evaluated using amino acids such as *L*-phenylalanine (**108**) and clay minerals or alumina. In this sense, the protocol allowed for obtaining a broad scope consisting of 24 examples of succinimides **109**, reaching yields of up to 99% and excellent *ee* (Scheme 38). In the optimization study, other non-biobased solvents were evaluated and the authors verified that diethyl ether (Et$_2$O) was the most effective, presenting conversion of >99% and 97% of *ee*.

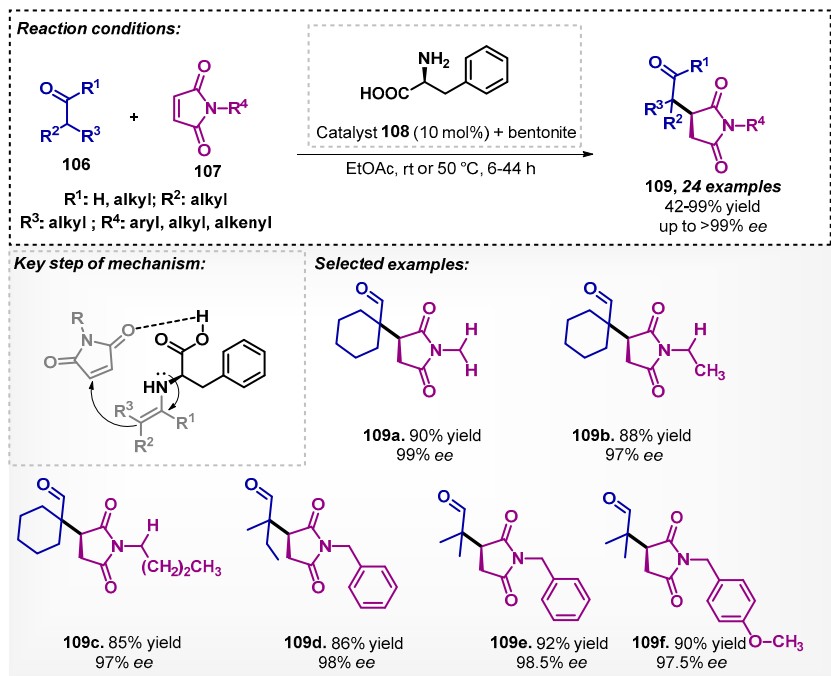

**Scheme 37.** Enantioselective aldol reaction of *α*-branched aldehydes **1** and glyoxylate derivatives **103**.

**Scheme 38.** Synthesis of succinimides **109** via asymmetric Michael reaction.

Jia and colleagues [85] designed a new catalyst based on Jiaphos with a diastereoisomeric P-chirogenic phosphine **112**. They performed the synthesis of the catalyst in five steps and evaluated its application in enantioselective (4+2) annulations. Several solvents were screened (toluene, acetonitrile, CHCl₃, THF, and dioxane), but the authors chose EtOAc due to the green chemistry point of view. Moreover, the chemoselectivity was improved by adding benzoic acid. Thus, under the optimized conditions, it was possible to synthesize 29 3,3′-spirocyclic oxindoles **113** via enantioselective allene–olefin (4+2) annulation with excellent yields and enantioselectivity (Scheme 39).

**Scheme 39.** Asymmetric synthesis of 3,3′-spirocyclic oxindoles mediated by catalyst **112**.

Syu and colleagues [86] studied an asymmetric Michael addition between ketones **6** and alkylidene malonates **114** mediated by a proline derivative **115** as organocatalyst (Scheme 40). The method allowed the preparation of 10 compounds **116** with low to excellent yields and good to excellent enantio- and diastereoselectivity. Due to the mild reaction conditions, long reaction times were necessary to obtain the compounds, ranging from 3 to 14 days. The authors proposed a transition state in which the enamine attacks the alkylidene malonates by the *Re*-face. In this protocol, other solvents were evaluated, and among them, the one that proved to be the most effective was THF, furnishing the desired product **116c** with 89% yield, 90:10 *dr,* and 94% *ee.*

Nakashima and Yamamoto [87] studied a flow-through chiral aldol reaction via a packed bed column with 5-(2-pyrrolidinyl)-1*H*-tetrazole (**117**) at room temperature. The authors evaluated different parameters, among them flow rate and catalyst column reactor volume (methods **A**, **B** or **C**) (Scheme 41). In general, seven compounds **8** were obtained by method **A** with yields between 26–99% and enantiomeric excesses reaching up to 97%. On the other hand, with method **B,** it was noticed that in most examples, the yields were ranged from 49–92%, whereas by method **C**, only one example was evaluated, being **8t** obtained with 86% yield and 62% *ee.* Furthermore, the authors expanded the method, replacing acetone (**16**) with cyclohexanone, and by using acetonitrile as solvent, the corresponding aldol product was obtained with 95% yield and 92% *ee.*

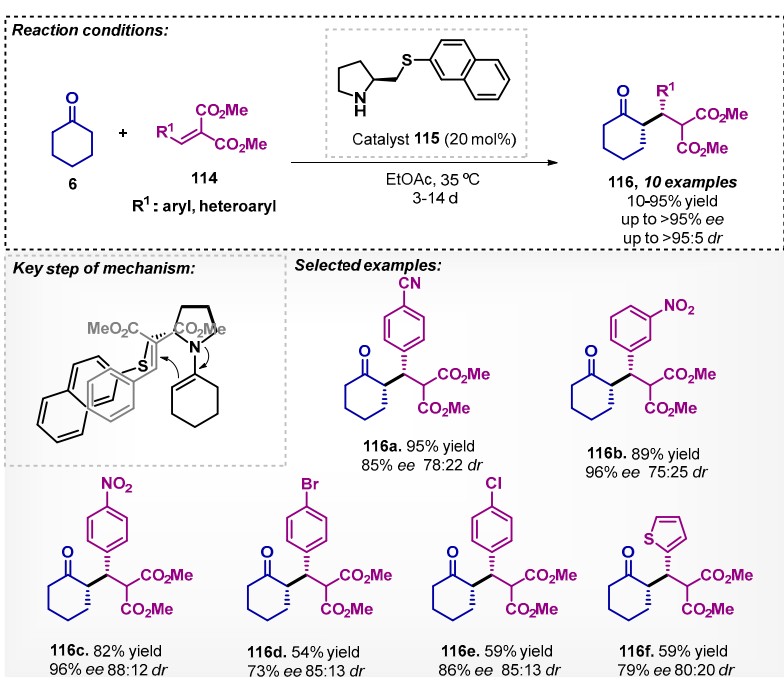

**Scheme 40.** Asymmetric Michael addition mediated by organocatalyst **115**.

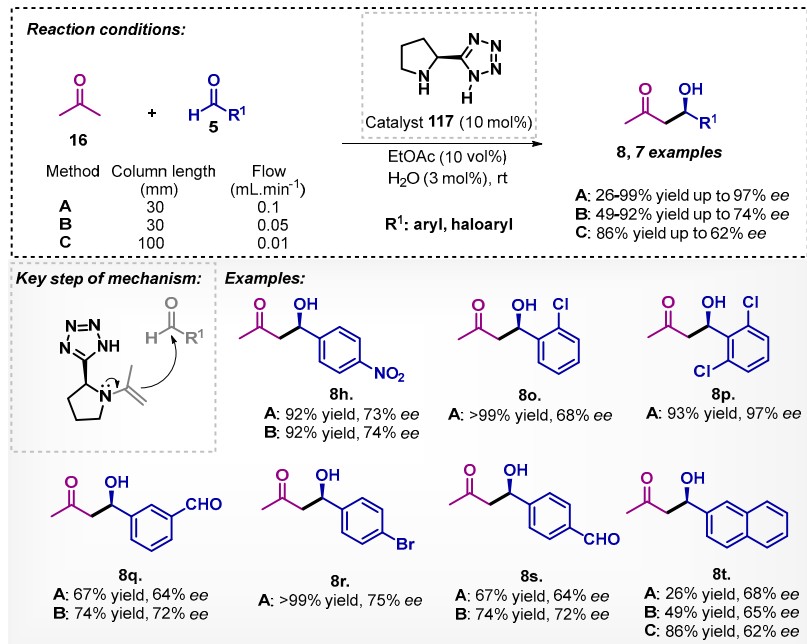

**Scheme 41.** Aldolic reaction catalyzed by proline-derived tetrazole in a flow system.

Monteiro and colleagues [88] investigated the synthesis of protected chiral aziridines **118** through the reaction between α,β-unsaturated aldehydes **20** and a protected amine **33** with the use of organocatalyst **21** (Scheme 42). The described protocol furnished five derivatives of protected chiral aziridines **118** with yields of 55–85%, *ee* reaching up to 85%, and a diastereoisomeric ratio of up to 92:8. In this study, a screening of solvents for the synthesis of aziridines **118** was carried out, and it was noted that $CH_3CN$ was the most promising solvent among the non-biobased ones but presented lower results in yield and *ee*. The authors used the chiral aziridines for the construction of reduced hydantoins via [3+2] annulation reaction with isocyanates.

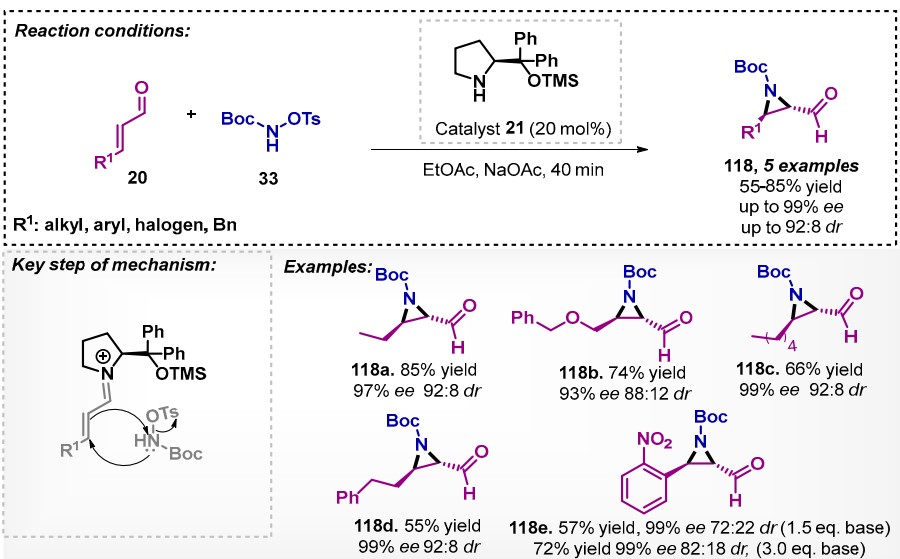

**Scheme 42.** Synthesis of protected chiral aziridines mediated by organocatalyst **21**.

Looking forward to the preparation of enantiomerically pure compounds that contain hydroxyphthalides **121** and **122**, Liu and colleagues [89] developed a new method based on *N*-heterocyclic carbene (NHC) obtained in situ by deprotonation of the triazolium salt **120**. The process consists of the formation of chiral acyl azolium intermediate from the reaction of aldehyde **5** and **120** in presence of 3,3′,5,5′-tetra-*tert*-butyldiphenoquinone (DQ) as oxidant. The authors also described examples using enals **20** without the need for oxidants to produce **122**. The reaction medium applies ethyl acetate as a solvent and *N,N*-diisopropylethylamine (DIPEA) as the base to obtain a series of derivatives with moderate to good *ee* (Scheme 43). It stands out that the authors also tested THF as solvent, but it showed a lower yield for **121a** (81%).

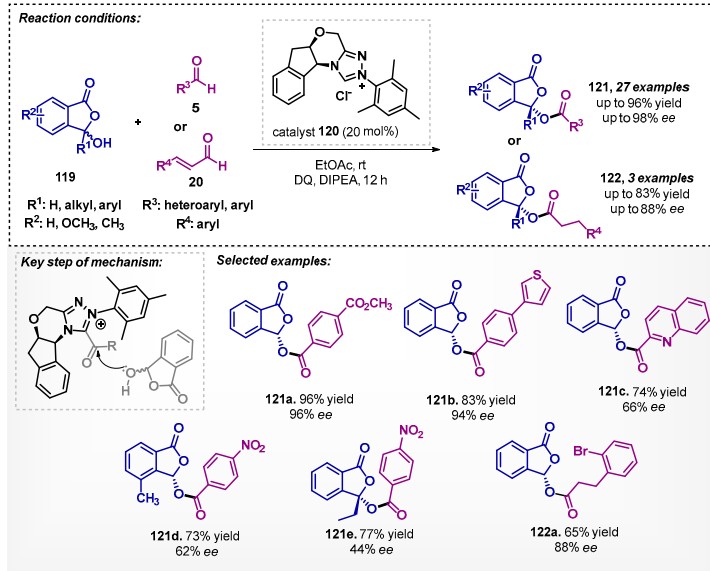

**Scheme 43.** Asymmetric acylation of hydroxyphthalides catalyzed by NHC derived from triazolium salt **120**.

In 2021, Ye's group [90] studied the use of the triazolium salt **124** in dynamic kinetic resolution (DKR) of α-trifluoromethyl hemiaminals **123**. They developed an efficient method to obtain asymmetric esters **125** in high yields and enantioselectivities up to 97% employing ethyl acetate as solvent, potassium carbonate as base, and 3,3′,5,5′-tetra-*tert*-

butyldiphenoquinone (DQ) as oxidant. Furthermore, THF, DCE, Et$_2$O, and EtOAc were also tested in the solvent screening, and ethyl acetate showed the best performance (92% yield and 93% *ee*), followed by THF (92% yield and 89% *ee*). It stands out that the authors carried out the synthesis of **125a** in a gram scale, reaching 99% yield with 94% *ee* (Scheme 44).

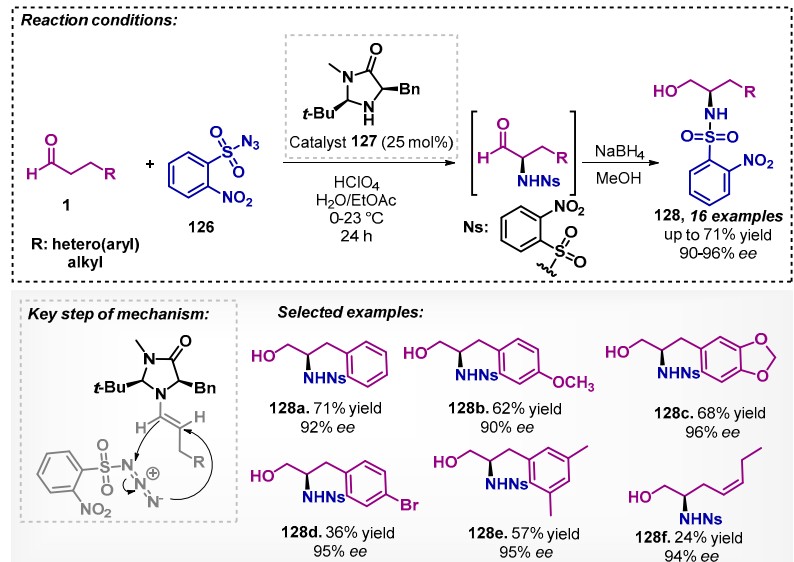

**Scheme 44.** Asymmetric acylation of hemiaminals **123** catalyzed by NHC derived from triazolium salt **124**.

McGorry and colleagues [91] described in their work a simple process to obtain a series of chiral amino alcohols **128** with high *ee*. They applied the MacMillan's second-generation imidazolidinone **127** to catalyze the α-sulfamidation of unbranched aldehyde **1** using sulfonyl azides **126** in aqua media with the addition of ethyl acetate (3–7% *v/v*), affording the corresponding α-amino aldehydes, which were then reduced with NaBH$_4$ (Scheme 45). During the optimization study, several solvents were evaluated, including DCM, toluene, THF, and acetone, but the best results were obtained with the H$_2$O/EtOAc mixture.

**Scheme 45.** α-Sulfamidation of unbranched aldehydes catalyzed by imidazolidinone **127**.

Wallbaum and colleagues [92] screened a series of imidazolidinone organocatalysts to promote the 1,3-chlorochalcogenation of meso-cyclopropyl carbaldehydes **130** with sulfenyl and phenylselenyl chlorides **129**. The best enantioselectivity was achieved with catalyst **133** using ethyl acetate as solvent at −4 °C, although better yield and diastereoselectivity were obtained with DCM. The enantioselective desymmetrization reaction proceeds first by forming the iminium and enamine intermediates to obtain products **132** after reduction with NaBH$_4$ on moderate to good yields, *ee* and diastereoselectivities (Scheme 46).

**Scheme 46.** Enantioselective desymmetrization of aldehydes **130** with catalyst **131**.

The potential applicability of ethyl acetate in organocatalyzed reaction was demonstrated by Yao and Wang [93] in 2014, employing the commercially available organocatalyst (DHQD)$_2$PHAL **135** for the *N*-allylic alkylation of hydrazone **133** with Morita–Baylis–Hillman (MBH) carboxylates **134**. The adducts **136** could be obtained with yields up to 94% and high *ee*. The same protocol, when evaluated in DCM, was inferior in terms of yields and stereoselectivities. The proposed transition state involves the formation of a more reactive intermediate via covalent bonding by quinuclidine attack on the MBH carboxylate (Scheme 47).

### 2.3.2. Activation Via Non-Covalent Bonding

Xie and colleagues [94] reported the enantioselective synthesis of spiro[indoline- 3,4′-pyrano[2,3-*c*]pyrazole] derivatives **139** using the commercially available catalyst (DHQD)$_2$PYR **138** with excellent yields and moderate to excellent enantioselectivities. During the optimization study, several solvents were evaluated (toluene, xylenes, CH$_2$Cl$_2$, CHCl$_3$, Et$_2$O, THF, acetonitrile, EtOH, DMF), and ethyl acetate was selected. The reaction occurs via organocatalytic asymmetric Michael/cyclization cascade reaction of *N*-benzyl isatylidene malononitriles **64** with 1-phenyl-3-methyl-5-pyrazolones **137**. The catalyst provides enantiocontrol through the formation of pyrazolone enolate, which, after its addition to isatylidene malononitrile, leads to the formation of the electrophilic intermediate, activated via hydrogen bonding of the protonated quinuclidine. The authors further derivatized product **139d** to a pyranotriazolopyrimidine, known for its biological activity (Scheme 48).

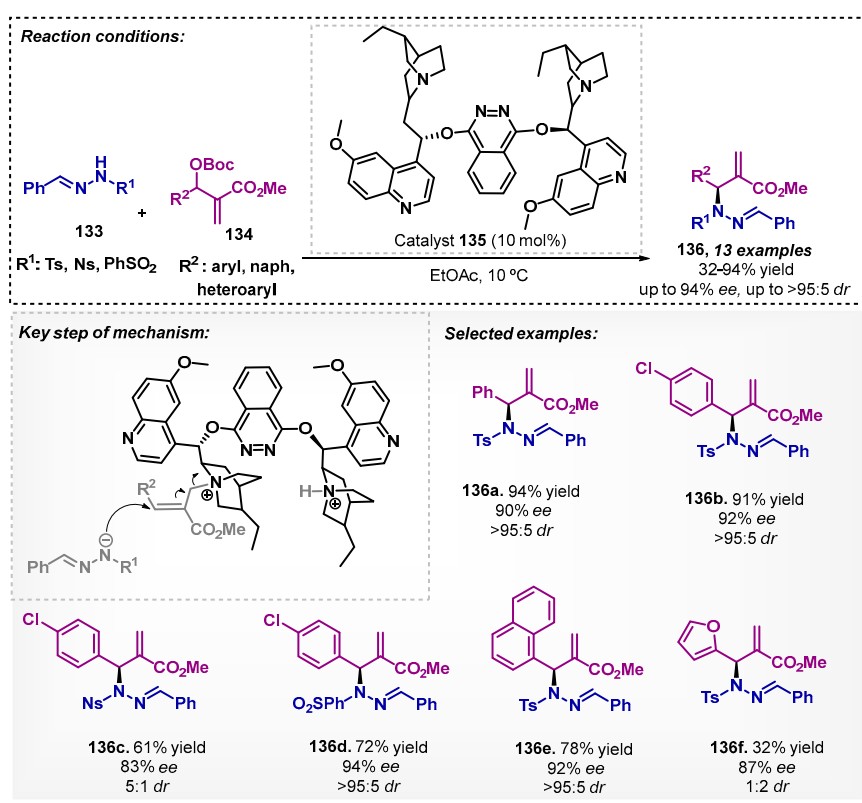

**Scheme 47.** Asymmetric *N*-allylic alkylation of hydrazone **133** with MBH carboxylates **134**.

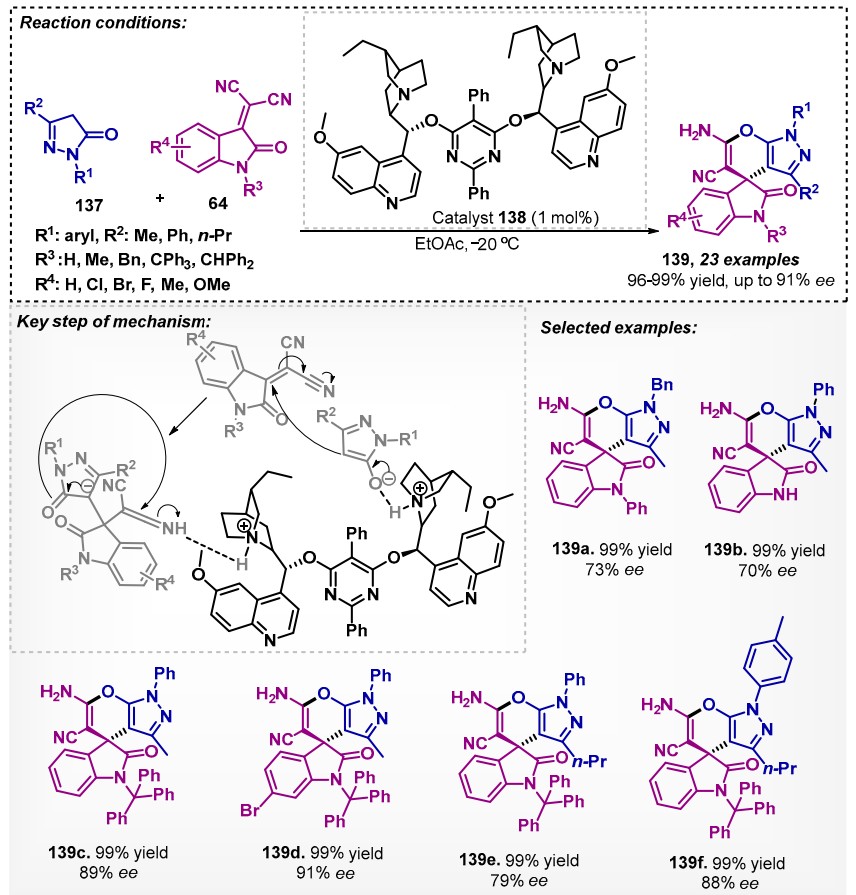

**Scheme 48.** Michael/cyclization cascade reaction of malononitrile **64** with pyrazolone **137**.

Ding and Wolf [95] demonstrated in their work an efficient asymmetric synthesis of *α*-oxetanyl and *α*-azetidinyl derivatives, obtained by reaction between oxindole **140** and nitroalkene **141** using ethyl acetate as a solvent and squaramide **142** as catalyst, providing the desired products with satisfactory yields and enantioselectivities (Scheme 49). In addition, they also performed reactions with acyclic *α*-fluoroketones **144** and **145** using the same conditions but in low to moderate *ee*. To improve the stereoselectivity, an optimization study was carried out providing cyclic and acyclic fluoroenolate asymmetric additions with reaction times between 70 to 110 h (Scheme 50).

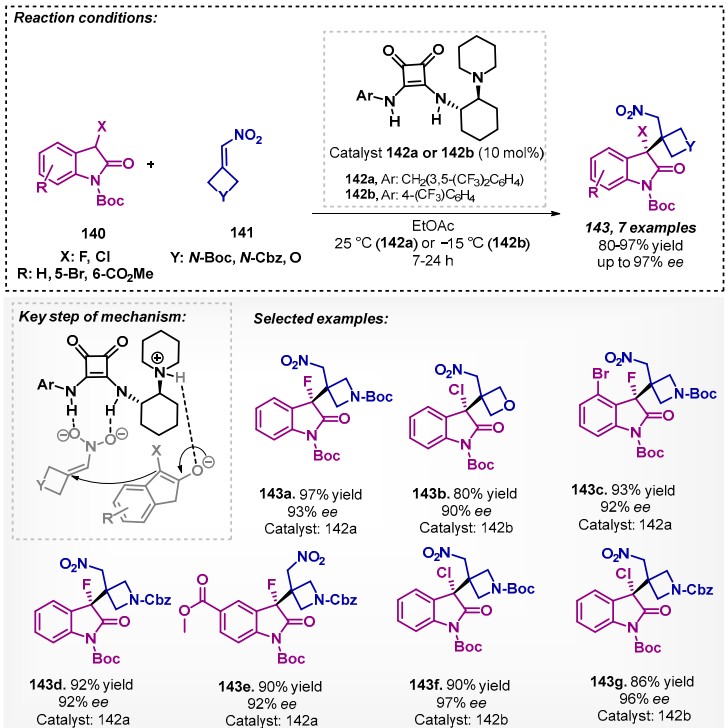

**Scheme 49.** Synthesis of *α*-oxetanyl and *α*-azetidinyl derivatives catalyzed by squaramides **142a,b**.

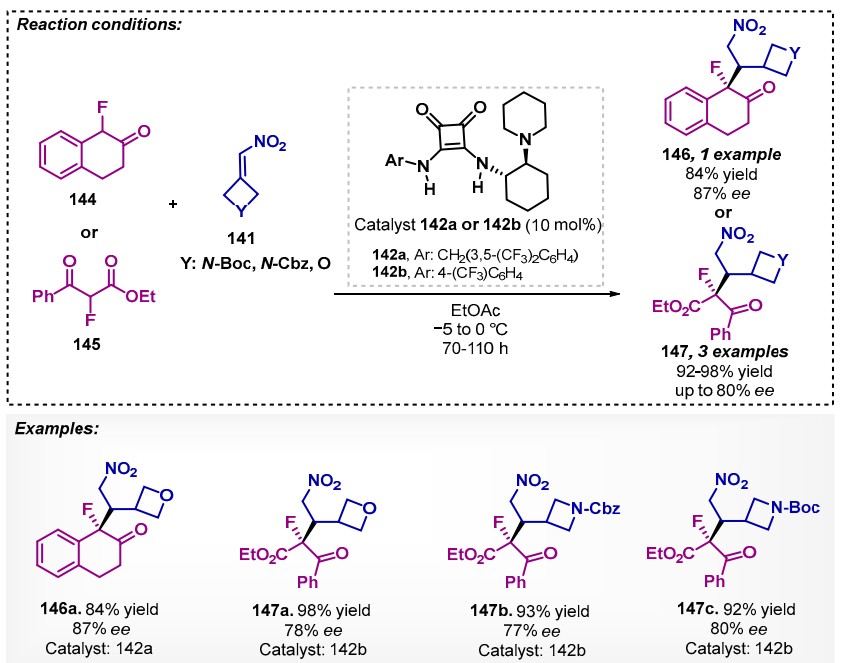

**Scheme 50.** Asymmetric fluoroenolate additions catalyzed by squaramides **142a,b**.

Moreover, the efficient use of squaramide analogue **150** as a chiral catalyst was demonstrated in the presence of ethyl acetate as solvent by Jang and coworkers [96]. They developed a method to promote the enantioselective addition of diphenyl phosphonate **148** to ketimines derived from isatins **149** with good to high yields and good to excellent enantioselectivities, except for compound **151b** (Scheme 51). In the optimization study, dichloromethane, chloroform, toluene, methanol, and tetrahydrofuran were also tested, but they showed a decrease in enantioselectivities when compared to EtOAc. It is worth noting that the synthetic procedure was tested on a gram scale to obtain **151a** with 81% yield and 93% *ee*.

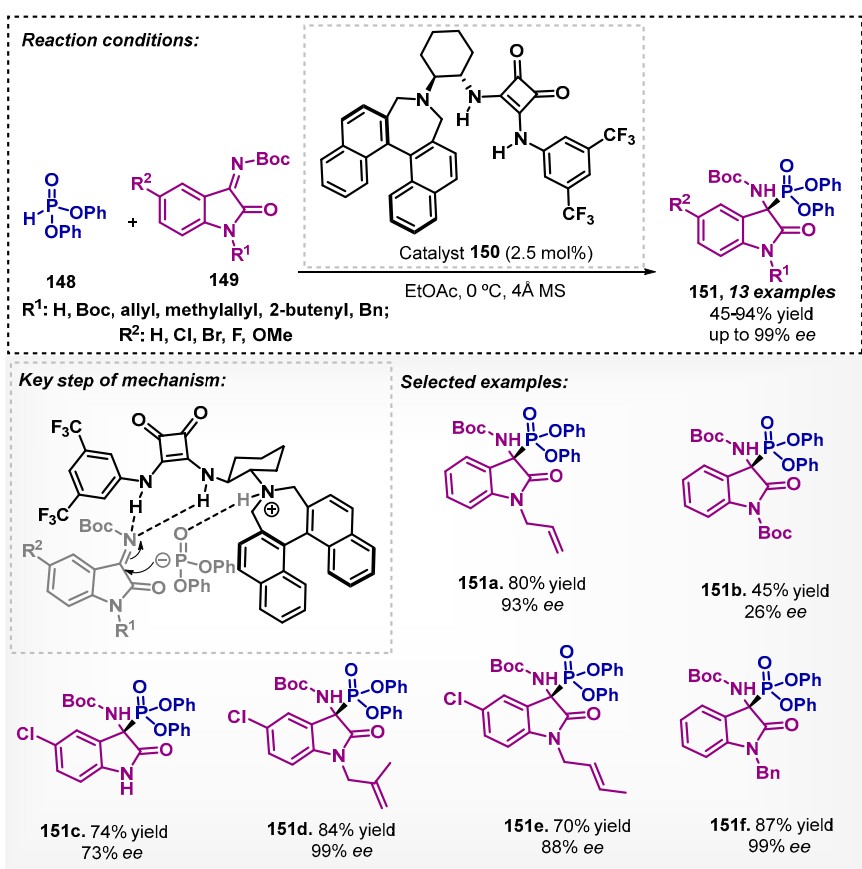

**Scheme 51.** Squaramide **150** promoted addition of diphenyl phosphonate to ketimines.

Interestingly, Wu and colleagues [97] developed a novel chiral bifunctional phosphine organocatalyst **152** bearing squaramide that was applied in the MBH reaction to obtain derived oxindoles, using isatins **15** and enone esters **58** as starting materials and ethyl acetate as solvent. The squaramide acts as a hydrogen bonding donor in the in-situ-generated phosphine enolate, which resulted in products **153** with moderately to excellent enantioselectivities and yields up to 99% (Scheme 52). It is noteworthy that MBH reactions are one of the main C-C bonding formation tools with excellent atom economy. Furthermore, they are useful tools in the formation of densely functionalized alcohols. This class of reactions is conventionally reported in DCM [98], THF [99], and chloroform [100] as solvents.

Du's group [101] carried out an asymmetric addition of 1,4-Michael between azadienes **154** and thiocyanoindanones **155**, mediated by a bifunctional organocatalyst **156** derived from squaramide. In this study, several solvents were evaluated, in which dichloromethane was the one that furnished the compound of interest with the highest yield, but when the authors used ethyl acetate, there was an improvement in enantio- and diastereoselectivities, so it was fixed as a solvent. Toluene also presented good results, with yields of up to 88% (slightly higher than EtOAc) but with lower *ee*, reaching 62%. Using the optimized condi-

tions, the method allowed for obtaining a scope consisting of 21 examples of compounds **157**, with yields of up to 87% and good enantio- and high diastereoselectivities (Scheme 53).

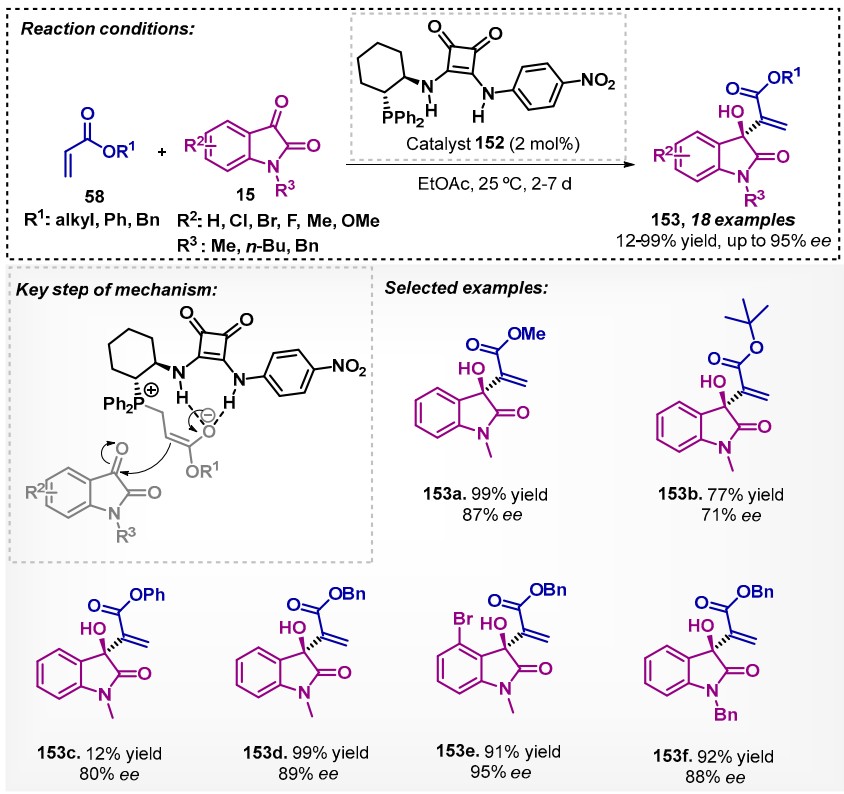

**Scheme 52.** Phosphine-squaramide **152** catalyzed MBH reaction.

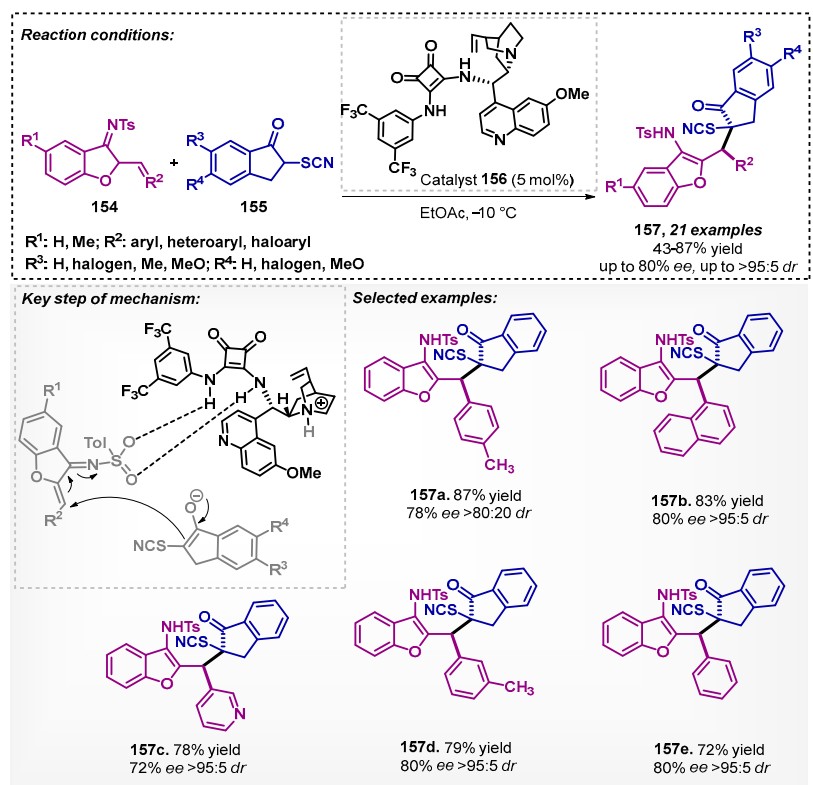

**Scheme 53.** Asymmetric 1,4-Michael reaction mediated by catalyst **156**.

Song and Du [102] described the enantioselective synthesis of 4-acyloxythiazole derivatives via organocatalytic cascade Michael/hemiketalization/retro-aldol reaction. To this end, the authors employed thiazolones **158** and α-nitroketones **159** as substrates with the bifunctional squaramide catalyst **142a** in the presence of ethyl acetate as solvent. The proposed mechanism pathway includes an activation of both substrates through hydrogen bonding. This method resulted in a wide variety of β-functionalized nitrocompounds **160** with moderate to excellent yields and up to 95% *ee*, even on a gram scale (Scheme 54).

**Scheme 54.** Asymmetric synthesis of 4-acyloxythiazole derivatives with squaramide catalyst **142a**.

Wu and co-workers [103] investigated the preparation of the synthetically useful and medicinally important 2*H*-thiopyrano [2,3-*b*]quinolones **162**, employing a squaramide-cinchone derivative **156** as catalyst. An asymmetric tandem Michael–Henry reaction in ethyl acetate between 2-mercaptoquinoline-3-carbaldehydes **161** to *trans*-nitrostyrenes **2** resulted in a protocol with excellent yields and stereocontrol. The proposed mechanism involves both deprotonation of the thiol and activation of nitrostyrene via hydrogen bonding by the catalyst (Scheme 55). Ethyl acetate as solvent proved to be important in the yield and enantioselectivity, since when the reaction was carried out with DCM, the product **162a** was obtained at 80% yield and 87% *ee*.

More recently, the utility of bifunctional squaramide **164**, in the synthesis of heterocyclic compounds, was further demonstrated by Shikari and colleagues [104] with spirooxindoles **165** via a domino reaction between isatin **15** with different *N*-substituents and γ-hydroxy enones **163**. During the solvent screening, toluene, 5, THF, and CH₃CN were evaluated, and EtOAc showed the best results. A wide variety of compounds were obtained with yields up to 90%, high diastereoselectivity, and excellent *ee*, including scale-up. The proposed transition state of this transformation involves increased electrophilicity of isatin by hydrogen bonding formed with the catalyst **164** (Scheme 56).

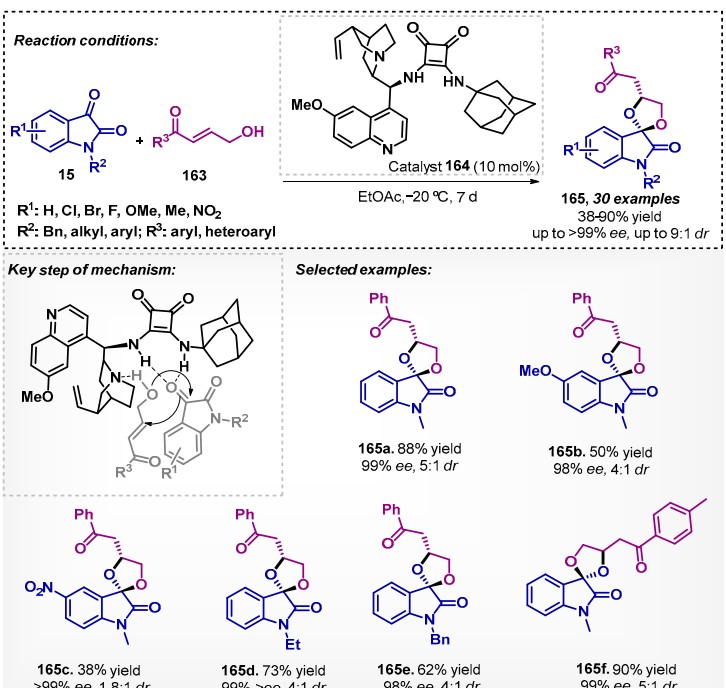

**Scheme 55.** Asymmetric tandem Michael–Henry reaction between 2-mercaptoquinoline-3-carbaldehydes **161** to *trans*-nitrostyrenes **2**.

**Scheme 56.** Asymmetric reaction between *N*-substituted isatins **15** and γ-hydroxy enones **163**.

Nagy and co-workers [105] reported the synthesis of binaphthyl-cinchone thio- and squaramides. The authors extend the application of these new organocatalysts into three reaction classes, among them, the enantioselective Michael addition reaction of acetylacetone **166** to *trans*-nitrostyrene **2**, affording products **168** with excellent yields and *ee* when using the organocatalysts **167a,b**. The best results were obtained in the presence of ethyl acetate when compared with THF as solvent. The reaction probably proceeded through

the activation of a squaramide hydrogen bonding donor with the nitrostyrene substrate, concomitantly with the formation of the enolate from quinuclidine moiety (Scheme 57).

**Scheme 57.** Enantioselective Michael addition reaction of acetylacetone **166** to *trans*-nitrostyrene **2**.

Zhang's group [106] studied the enantioselective synthesis of dihydropyrimidinethiones (DHPM) **172** mediated by a bisphosphorylimide type organocatalyst **171** (Scheme 58). Under the optimized condition, which included a reaction time of 12 h at 50 °C, it was possible to obtain 13 examples of DHPM **172** with yields between 83–97% and good *ee*. The method was developed via a Biginelli multicomponent reaction, which stands out for its convergence and atom economy. In this study, other solvents were evaluated and, among them, dichloromethane proved to be efficient, affording the product **172a** with 75% yield and 91% *ee*.

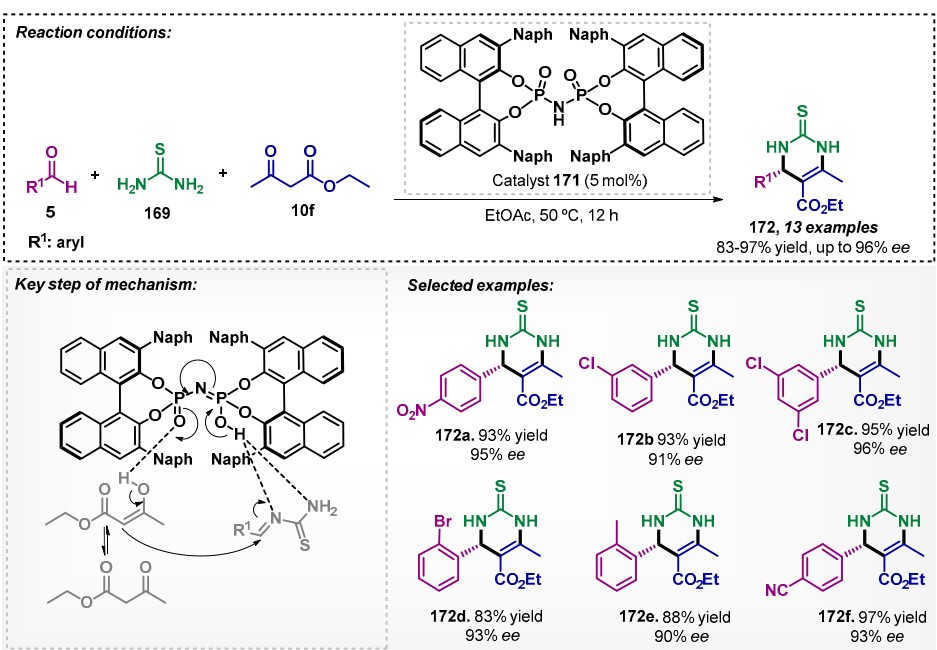

**Scheme 58.** Bisphosphorylimide **171** mediated asymmetric Biginelli reaction.

Liu and colleagues [107] used a chiral phosphoric acid (CPA) **175** as catalyst in ethyl acetate as solvent for the asymmetric allylation of isatin-derived 3-indolylmethanols **174**. The phosphoric acid catalyst activated simultaneously both substrates via hydrogen bonding,

which allows a cascade reaction, i.e., the addition of *o*-hydroxystyrenes **173** followed by an elimination of hydrogen resulting in compounds **176** with excellent diastereo- and enantio-control and low to good yields (Scheme 59). This reaction showed the best stereoselectivity with EtOAc when compared with other solvents such as toluene and dichloroethane.

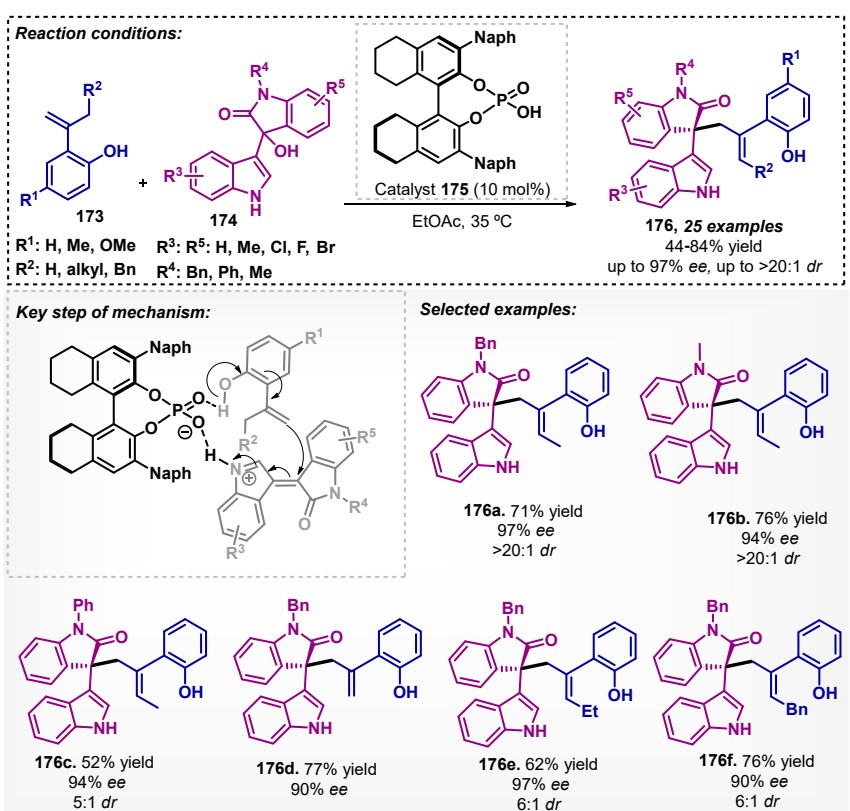

**Scheme 59.** Allylation of 3-indolylmethanols **174** with *o*-hydroxystyrenes **173**.

Later, the same class of catalyst was reported by Shao and Cheng [108], where an investigation was conducted with indolenines **178** and sulfones **177**. The synthesis of benzothiazole derivatives **180** with good yield and enantioselectivity occurred through a cascade Mannich addition/Smiles rearrangement (Scheme 60). Although Mannich reactions are conventionally described in chloroform [109] and THF [110], in the presence of ethyl acetate, the reaction demonstrated to be more efficient than in DCE, toluene, and THF.

More recently, Shao and colleagues [111] conducted studies on the asymmetric aza-Henry reaction promoted by the cinchona-thiourea organocatalyst **181** via hydrogen bonding. In this sense, the reaction of indolenines **178** with nitromethane **39** resulted in adducts **182** with up to 81% yield and 94% *ee* (Scheme 61). In the solvent screening, ethyl acetate proved to be more efficient than DCM and THF.

Rodriguez's group [112] studied the production of chiral polysubstituted cyclohexanes **185** using ketoamides **183** and nitroalkenes **2**, mediated by the Takemoto catalyst (derived from thiourea) **184**. The reaction was carried at room temperature for four days. Under the optimized protocol, it was possible to obtain five examples of **185**, with yields of up to 67% and up to 98% *ee* and good diastereoselectivities (Scheme 62). During the study, other solvents were evaluated, and it was noted that the most effective was DCM, which led to the formation of the product with 45% yield, although much lower than that obtained with EtOAc. The authors also addressed the reaction between 1,3-ketoamides with acrolein by using other non-bio-based solvents.

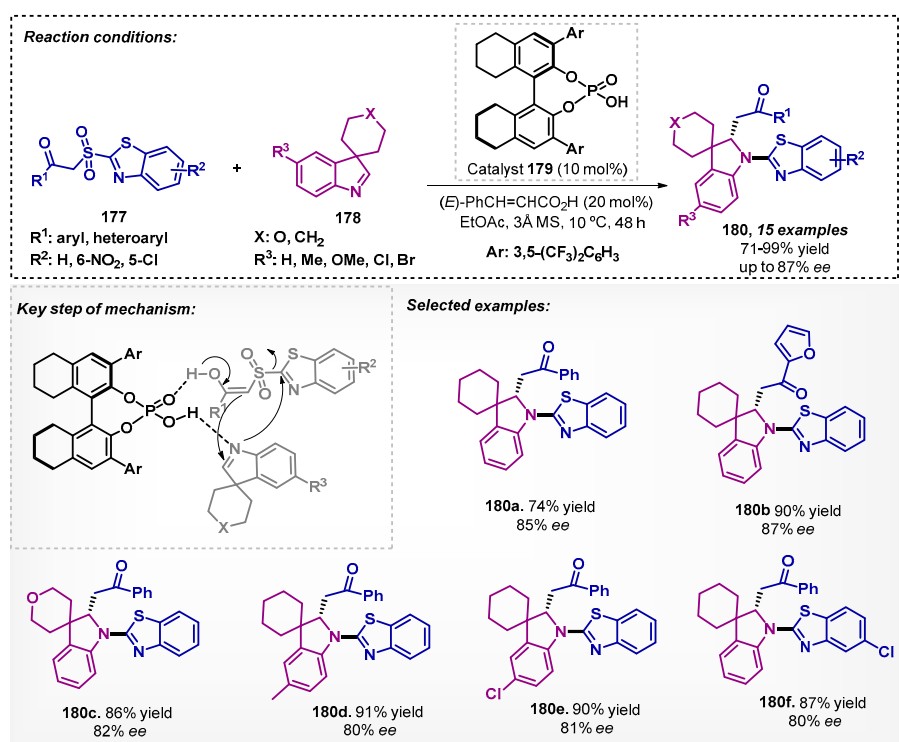

**Scheme 60.** Cascade Mannich addition/Smiles rearrangement of indolenines **178** and sulfones **177**.

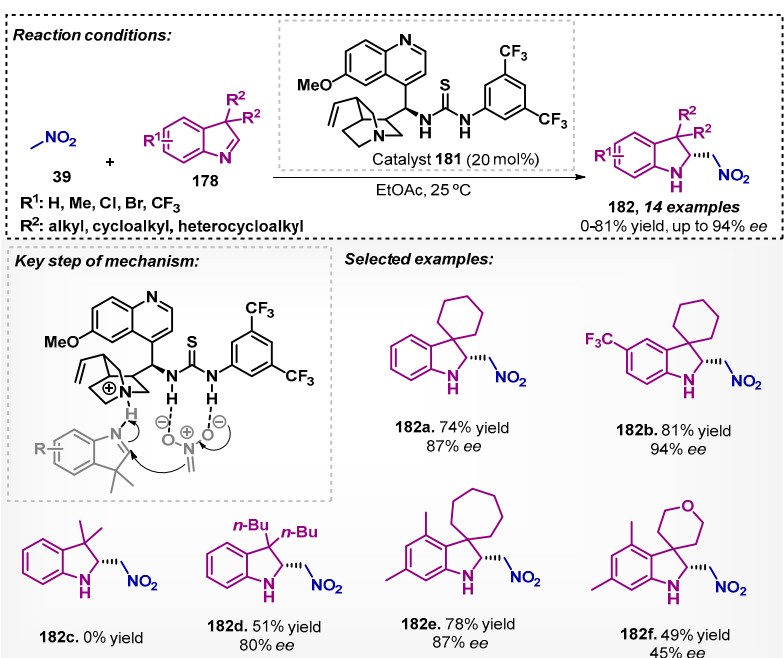

**Scheme 61.** Asymmetric aza-Henry reaction of indolenines **178** to nitromethane **39**.

Kumar and colleagues [113] demonstrated the synthesis of α-aminophosphonates in the presence of ethyl acetate as solvent, employing a cinchona-derived thiourea **186** as chiral organocatalyst. Using a simple protocol, they were able to obtain 12 derivatives **151** with good yields and enantioselectivity. Interestingly, in the proposed transition state, the quinuclidine nitrogen in catalyst **186** plays an important role in the equilibrium between phosphite and phosphonate shifting to reactive phosphite form (Scheme 63). In this work, there was also a screening of solvents, and it was observed that xylene presented a satisfactory result in terms of yield (87%) but lower *ee* (72%) when compared with EtOAc.

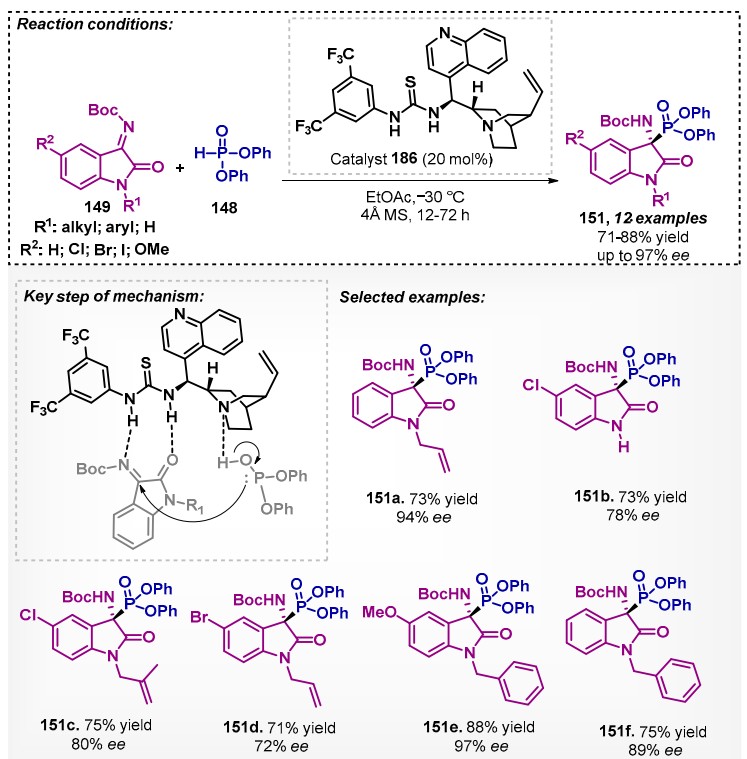

**Scheme 62.** Takemoto's catalyst mediated the synthesis of polysubstituted cyclohexanes **185**.

**Scheme 63.** Asymmetric synthesis of α-amino phosphonates catalyzed by thiourea derivative **186**.

Chen's group [114] developed an organocatalyzed formal [5+1] annulation using DABCO via a cascade Michael-aldol reaction in ethanol resulting in products in racemic form. To obtain enantiomerically enriched products, they tested this protocol using quinine derived thiourea **181** as a chiral catalyst and ethyl acetate as solvent to obtain **189** with 79% enantioselectivity and up to >95:5 *dr* (Scheme 64).

**Scheme 64.** Enantioselective [5+1] annulation catalyzed by quinine derived thiourea **181**.

## 2.4. Supercritical Carbon Dioxide

Carbon dioxide can be mainly obtained from renewable natural sources (plants) and industrial processes (burning of fuels) [115]. The $CO_2$ becomes a supercritical fluid when the temperature and pressure are higher than its critical values (31.1 °C and 7.38 MPa). The supercritical carbon dioxide (sc$CO_2$) is considered a green solvent since it is stable, widely available, non-toxic and potentially recyclable, and presents low viscosity and high diffusivity [116].

### 2.4.1. Activation Via Covalent Bonding

Liu's group evaluated the effect of compressed $CO_2$ on an asymmetric aldol reaction promoted by *L*-proline derivative **190** as catalyst [117]. They observed that compressed $CO_2$ induced the formation of supramolecular assemblies of amphiphilic proline derivative in water. The developed method proved to be efficient since it allowed for obtaining the compound of interest *ent*-**8g** with 99% yield and 93% *ee*. In the proposed mechanism, the bifunctional catalyst acts via a combination of hydrogen bonding and covalent activation (Scheme 65). The authors also evaluated the recovery and reuse of the organocatalyst, which remained active even after five reaction cycles.

**Scheme 65.** Aldol reaction assisted by vesicles regulated by compressed $CO_2$.

### 2.4.2. Activation Via Hydrogen Bonding

In 2012, Zlotin's group [118] described the first enantioselective Michael synthesis of malonates **9** or **35** to α-nitroolefines **2** using liquid $CO_2$ as the solvent and a thiourea derivative catalyst **184** (Scheme 66). The transition state involves the formation of a more reactive intermediate via hydrogen bonding by thiourea and deprotonation of the malonate by tertiary amine. The developed protocol resulted in the preparation of 13 compounds **81** with moderate to good *ee*. Although these results are below those obtained by Ashokkumar and Siva [70], the protocol allows for obtaining the same class of compounds without the need for structural modification of the catalyst and in the presence of liquefied $CO_2$, and the reaction was demonstrated to be similar to that in toluene with 60% yield and 92% *ee* for **81a**.

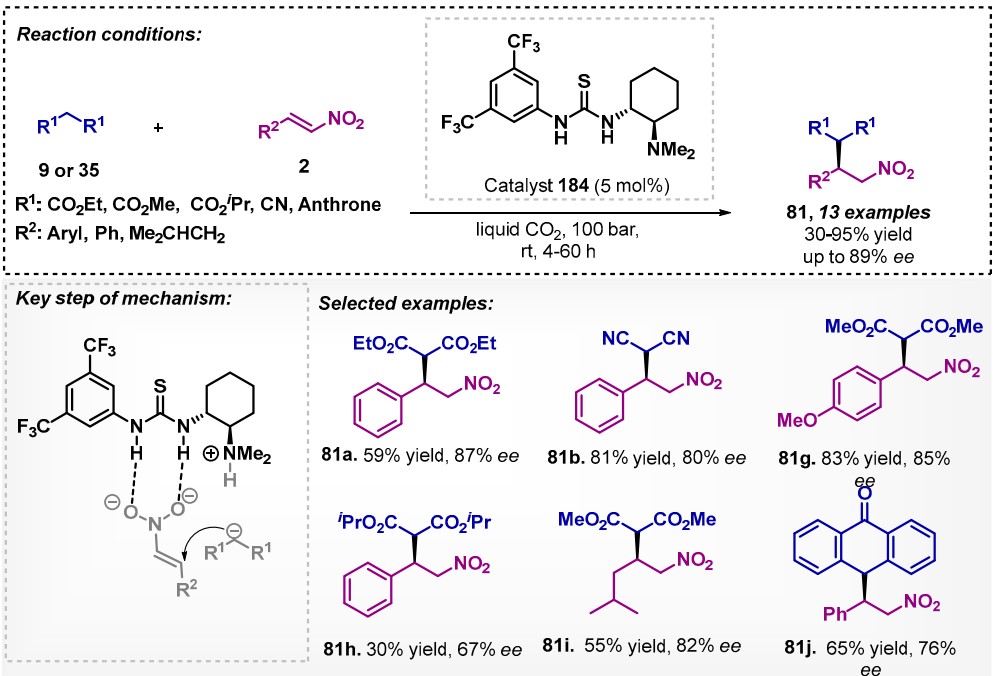

**Scheme 66.** Asymmetric Michael addition catalyzed by thiourea derivative **184**.

Kuchorov and colleagues studied an asymmetric version of the Michael addition in a supercritical carbon dioxide system using diphenylphosphite (**148**) and nitroalkenes **2** as starting materials [119]. In this protocol, the authors evaluated bifunctional organocatalysts **191** and **192** containing the tertiary amino group and the squaramide moiety. The method allowed for obtaining β-nitrophosphonates **193** with yields of up to 97% and 94% *ee*. The proposed transition state includes an activation of the nitroalkene through hydrogen bonding by the squaramide, followed by deprotonation of the phosphite by the tertiary amine (Scheme 67). Squaramide **191** bearing the adjacent piperidine unit showed the best catalytic performance in the model reaction in liq-$CO_2$, which was similar to or just slightly lower than the corresponding results obtained earlier in the $CH_2Cl_2$, toluene, or $Et_2O$ media.

Zlotin's group reports the first use of sc$CO_2$ as the solvent in enantioselective catalytic domino-reactions [120]. The authors demonstrated that chiral tertiary-amine squaramide **192** was able to catalyze the Michael reaction between chalcone **89** and α-nitroalkene **2** to obtain the desired products **194** with high yields (up to 97%) and enantioselectivity (up to 98% *ee*) (Scheme 68). Moreover, trifluoromethane ($CHF_3$) in a supercritical state was also tested, and the performance was similar to the sc$CO_2$.

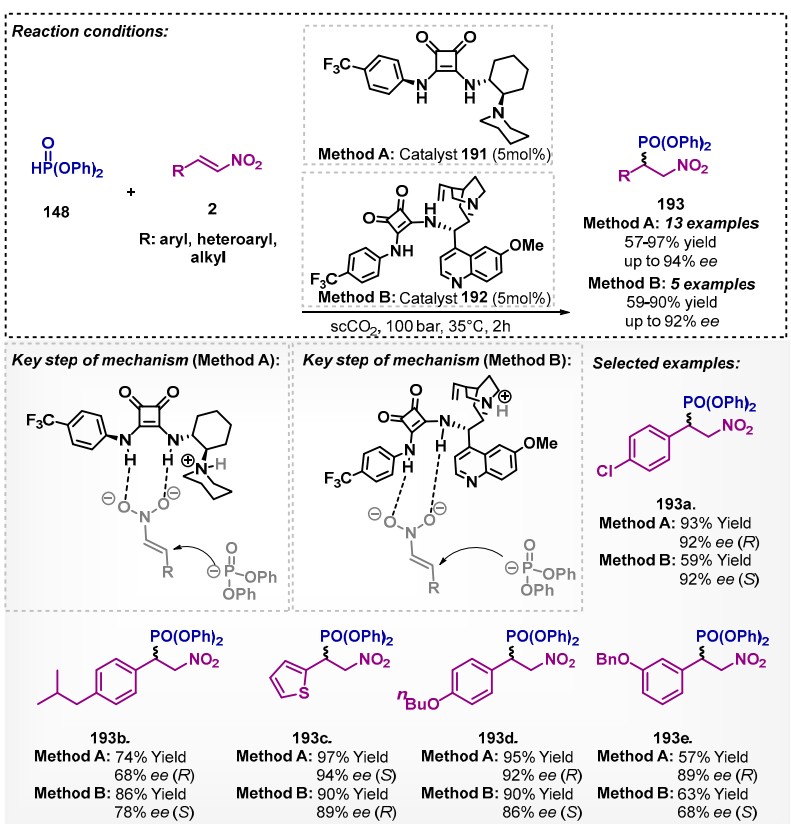

**Scheme 67.** Asymmetric Michael addition reaction in scCO$_2$.

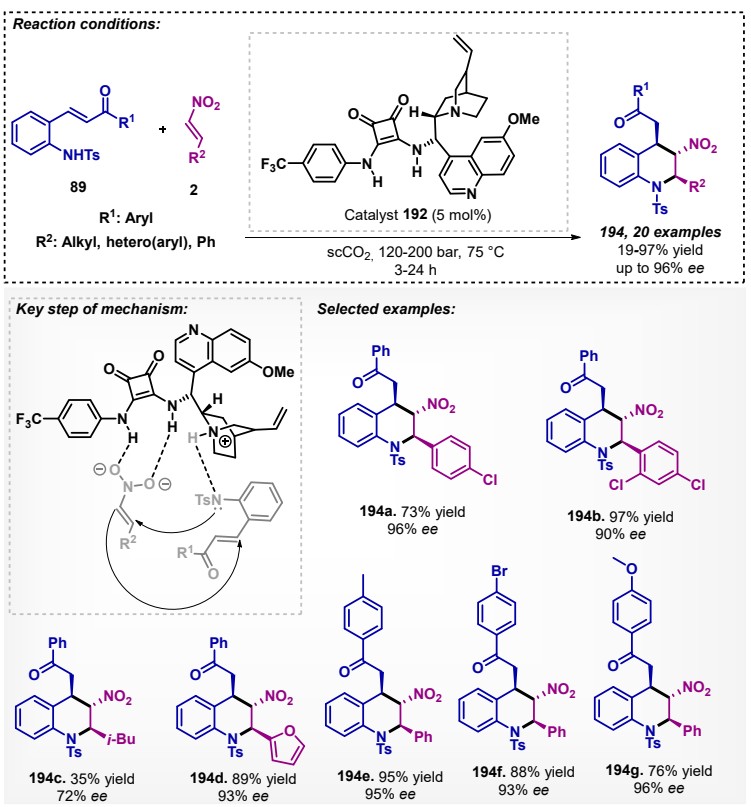

**Scheme 68.** Michael-initiated asymmetric domino-reactions catalyzed by squaramide **192**.

The same group described the application of bifunctional tertiary-amine squaramides in Michael reactions in the presence of scCO$_2$ as the solvent [115]. In this case, they developed the novel pseudo-enantiomeric bifunctional squaramides **195** and **196**, modified with the long-chained alkoxy group, and obtained the desired Michael adducts **194** and *ent*-**194** with moderate to high yields (up to 98%) and high enantioselectivity (up to 98% *ee*) (Scheme 69).

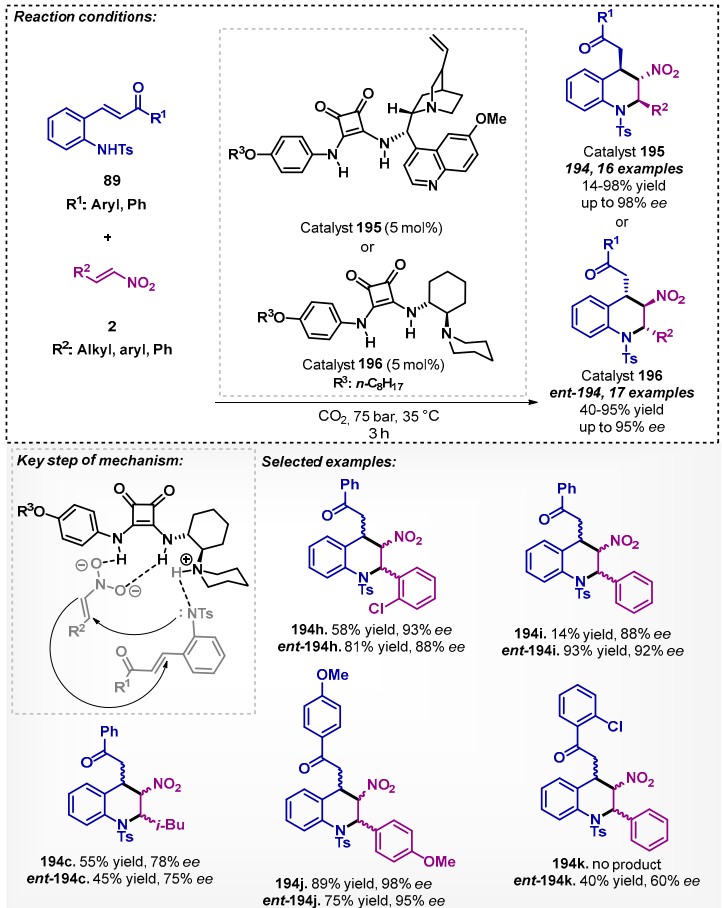

**Scheme 69.** Asymmetric double-Michael cascade catalyzed by squaramide **195** and **196**.

### 2.5. Ethyl Lactate

Ethyl lactate is a nontoxic and biodegradable solvent formed by the esterification reaction of ethanol and lactic acid that can be generated from biomass raw materials through fermentation [121]. This bio-based solvent is used in additives, perfumes, and as a cleaning agent, and its physical properties are adequate to replace methylene chloride and chloroform.

#### Activation Via Non-Covalent Bonding

As far as we know, only one example in asymmetric organocatalysis using ethyl lactate as a solvent has been described so far. In 2021, Verková and colleagues [122] developed a sequential one-pot two-step Michael addition applying squaramide **197** as catalyst. When the substrate was cyclohexanone **6**, a basic additive NaOAc and co-catalyst (*S*)-proline were required, affording the intermediate lactol **198**, and without any purification, the product **199** was obtained after sequential reductive etherification with 70% yield (66% *ee*, 95:5 *dr*) (Scheme 70).

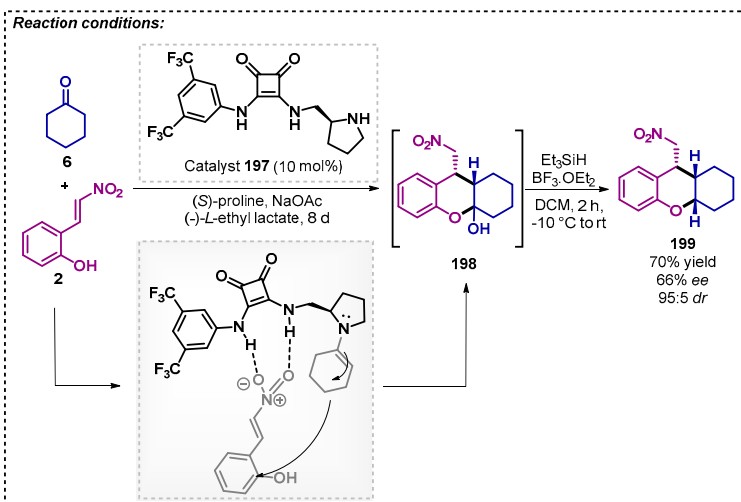

**Scheme 70.** Asymmetric Michael reaction catalyzed by squaramide **197** followed by reduction.

Interestingly, when the authors used a more reactive Michael donor substrate, such as methyl 2-oxocyclopentane-1-carboxylate **200**, neither the additive nor the co-catalyst were necessary, and in addition, a shorter reaction time was observed (3 d compared to 8 d). Thus, the product **202** was obtained with 72% yield and enantiomeric purity > 99%. (Scheme 71).

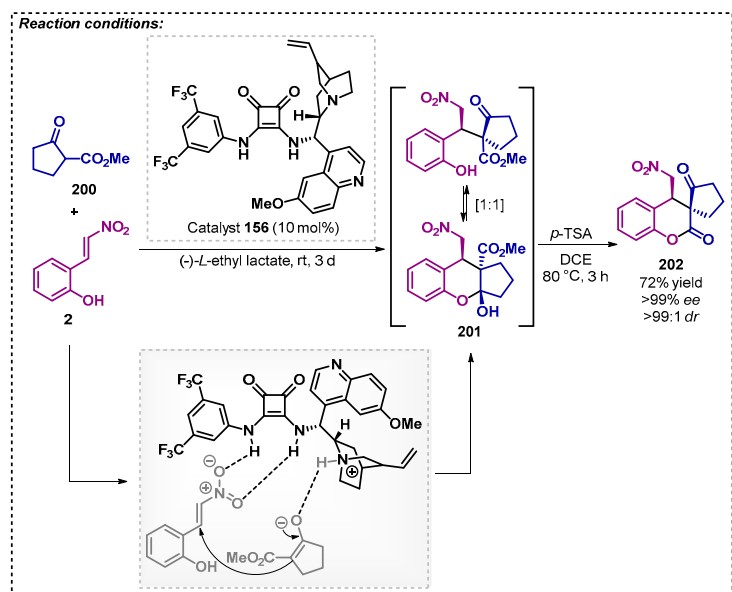

**Scheme 71.** Asymmetric Michael addition–cyclization catalyzed by squaramide **156**.

### 2.6. Diethyl Carbonate

The industrial production of monomeric dialkyl carbonates has historically used condensation of the corresponding alcohol with phosgene as a carbonyl source. However, the production of diethyl carbonate (DEC) has become more sustainable by being able to incorporate $CO_2$ into organic molecules [123]. This solvent has ecofriendly features, such as low toxicity, polarity, and high biodegradability.

Activation Via Non-Covalent Bonding

Bhanage and Ganesh described the synthesis of 1,2,3,4-tetrahydroquinolines (**206**) through organocatalyzed asymmetric hydrogenation [124]. In this method, substituted quinolines **203** were adopted as starting materials, Hantzsch esters (**204**) as the hydrogen

source, and the chiral Brønsted acid **205** was evaluated as catalyst. Propylene carbonate, dimethyl carbonate, and diethyl carbonate were screened as solvents, and the best results were obtained with DEC. In the proposed mechanism, the quinoline and Hantzsch ester (HEH) were activated via hydrogen bonding by the catalyst **205**. Under the optimized condition, this protocol allowed for obtaining nine examples of 1,2,3,4-tetrahydroquinolines **206** with excellent yields and *ee* (Scheme 72).

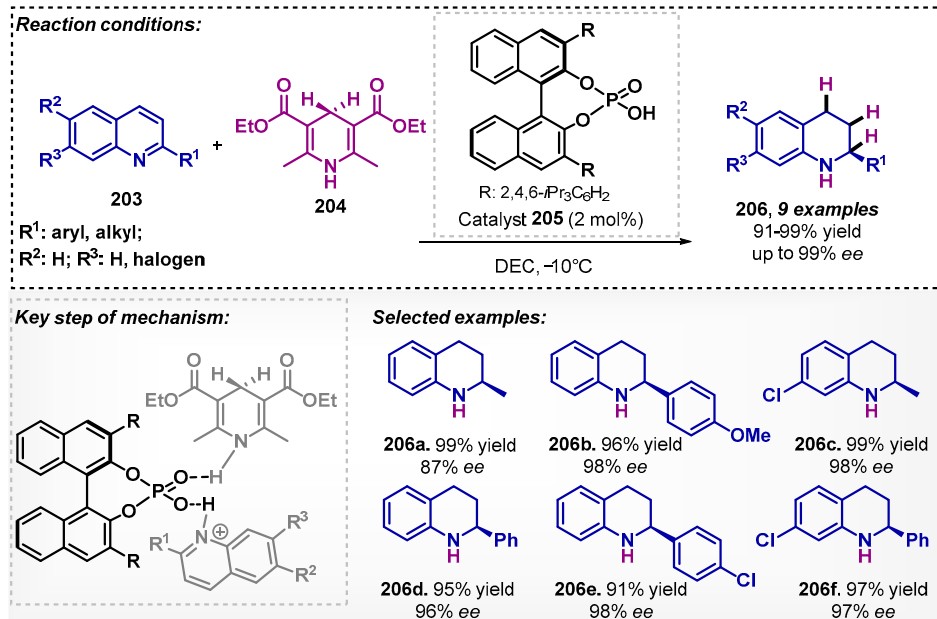

**Scheme 72.** Hydrogenation of quinolines catalyzed by chiral phosphoric acid.

### 3. Conclusions

In summary, in this review, we discussed the reports published in the last decade on asymmetric organocatalysis using bio-based solvents. A wide variety of proline, cinchona, and BINAP derivatives were shown to be effective catalysts, including in obtaining pharmacologically active compounds. Moreover, many important organic reactions, including the Morita–Baylis–Hillman, Mannich, cycloadditions, and especially, the Michael addition, may be efficiently carried out in their asymmetric version using bio-based solvents. Some methods also proved to be highly efficient in scale-up procedures by using heterogeneous catalysts in a continuous flow regime.

However, mainly ethanol, 2-MeTHF, ethyl acetate, and supercritical $CO_2$ have been explored so far, thus, further studies are needed to evaluate other bio-based solvents such as cyrene, $\gamma$-valerolactone, and limonene. The use and acceptance of these renewable solvents can be increased only if competitive prices are achieved to replace traditional organic solvents.

Despite the current growth in the development of more sustainable protocols, it is evident that most of the examples mentioned in this review demonstrated that the solvent choice was due to its greater efficiency in the process and not exclusively because of the environmental impact. However, it is not possible to generalize that the bio-based solvents are always the best option since small changes in the structure of substrates may modify significantly the solubility and, thus, the reactivity. As we have demonstrated, a large number of successful examples have already been reported in the literature, which should stimulate other research groups to consider these alternative greener solvents in the near future.

**Author Contributions:** Writing—original draft preparation, review and editing, L.S.R.M.; writing—original draft preparation, I.V.M.; writing—original draft preparation, J.R.N.d.S.; review, editing, supervision and funding acquisition, A.G.C. All authors have read and agreed to the published version of the manuscript.

**Funding:** The authors acknowledge Fundação de Amparo à Pesquisa do Estado de São Paulo (grants 2018/23761-0, 2013/07600-3, 2014/50249-8), Coordenação de Aperfeiçoamento de Pessoal de Nível Superior (finance code 001), GlaxoSmithKline and Conselho Nacional de Desenvolvimento Científico e Tecnológico (grants 429748/2018-3 and 302140/2019-0) for financial support and fellowships.

**Data Availability Statement:** Not applicable.

**Conflicts of Interest:** The authors declare no conflict of interest.

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
