# Peer review of "Recent Advances in Greener Asymmetric Organocatalysis Using Bio-Based Solvents"

_catalysts, doi:10.3390/catal13030553_

Round 1
Reviewer 1 Report
The authors aimed to show the recent literature on asymmetric organocatalysis using bio-based solvents. The authors have shown cases where bio-based solvents have proven more beneficial than conventional organic solvents. The authors have also identified the lack of evaluation for other bio-based solvents.
The manuscript is well structured, with relevant citations and clearly presented schemes. The authors reviewed asymmetric organocatalysis in ethanol, 2-MeTHF, and ethyl acetate. The authors have shown the importance and benefit of bio-based solvents over conventional organic solvents. Additionally, the authors have identified the lack of literature on other bio-based solvents in asymmetric organocatalysis.
Line 286: heterogeneous catalyst 48 should be changed to heterogeneous catalyst 50.
The authors should clearly state when the organocatalyst is derived from proline throughout the manuscript as opposed to stating proline.
I also don't think it's necessary to mention the solvent for each reaction since they are only discussing one solvent per a sub-heading (three solvents in total Ethanol, 2-MeTHF, Ethyl acetate) Also, see attached file for minor grammatical errors.

Author Response
The authors aimed to show the recent literature on asymmetric organocatalysis using bio-based solvents. The authors have shown cases where bio-based solvents have proven more beneficial than conventional organic solvents. The authors have also identified the lack of evaluation for other bio-based solvents.
The manuscript is well structured, with relevant citations and clearly presented schemes. The authors reviewed asymmetric organocatalysis in ethanol, 2-MeTHF, and ethyl acetate. The authors have shown the importance and benefit of bio-based solvents over conventional organic solvents. Additionally, the authors have identified the lack of literature on other bio-based solvents in asymmetric organocatalysis.
Thank you.
Line 286: heterogeneous catalyst 48 should be changed to heterogeneous catalyst 50.
OK
The authors should clearly state when the organocatalyst is derived from proline throughout the manuscript as opposed to stating proline.
This has been fixed.
I also don't think it's necessary to mention the solvent for each reaction since they are only discussing one solvent per a sub-heading (three solvents in total Ethanol, 2-MeTHF, Ethyl acetate) Also, see attached file for minor grammatical errors.
OK.
Reviewer 2 Report
Corrêa and co-workers report in this manuscript a comprehensive overview on the organocatalytic methodologies involving bio-based solvents. The manuscript is well organised and the discussion is divided in a logical order, depending on the type of organocatalytic activation as well as the used bio-based solvents. Figures and schemes are clear, consistent in style and depict clearly the most important examples, the results of each discussed paper and the reactions taking place. Moreover, the stereoselective transition states are adequately discussed and explained. This review covers a pivotal topic in modern organocatalysis, since the use of alternative solvents and greener approaches is surely a key challenge in asymmetric synthesis. This review could be of high interest for the readers of the journal. This reviewer believes that the manuscript is suitable for the journal scope and supports publication after few revisions.
In the introduction section, the state of the art of organocatalysis is explained, however, while the aspect of green solvents is well cited, more recent revews should be cited on organocatalysis development. I would suggest to add the following references:
- ChemSusChem, 2021, 2785 - https://chemistry-europe.onlinelibrary.wiley.com/doi/full/10.1002/cssc.202100573
- Nature Review Chemistry, 2019, 491 - https://www.nature.com/articles/s41570-019-0116-0
- Chem. Soc. Rev., 2017, 1661 - https://pubs.rsc.org/en/content/articlelanding/2017/CS/C6CS00757K#!divAbstract
- Chem. Rev., 2014, 8807 - https://pubs.acs.org/doi/full/10.1021/cr500235v
Author Response
Corrêa and co-workers report in this manuscript a comprehensive overview on the organocatalytic methodologies involving bio-based solvents. The manuscript is well organised and the discussion is divided in a logical order, depending on the type of organocatalytic activation as well as the used bio-based solvents. Figures and schemes are clear, consistent in style and depict clearly the most important examples, the results of each discussed paper and the reactions taking place. Moreover, the stereoselective transition states are adequately discussed and explained. This review covers a pivotal topic in modern organocatalysis, since the use of alternative solvents and greener approaches is surely a key challenge in asymmetric synthesis. This review could be of high interest for the readers of the journal. This reviewer believes that the manuscript is suitable for the journal scope and supports publication after few revisions.
Thank you.
In the introduction section, the state of the art of organocatalysis is explained, however, while the aspect of green solvents is well cited, more recent reviews should be cited on organocatalysis development. I would suggest to add the following references:
- ChemSusChem, 2021, 2785 - https://chemistry-europe.onlinelibrary.wiley.com/doi/full/10.1002/cssc.202100573
OK
- Nature Review Chemistry, 2019, 491 - https://www.nature.com/articles/s41570-019-0116-0 OK
- Chem. Soc. Rev., 2017, 1661 - https://pubs.rsc.org/en/content/articlelanding/2017/CS/C6CS00757K#!divAbstract
OK
- Chem. Rev., 2014, 8807 - https://pubs.acs.org/doi/full/10.1021/cr500235v
OK
We have also made all the corrections made by the referee in the manuscript pdf file.
Reviewer 3 Report
The review by Martelli et al. is dealing with an important topic, a superposition of popular areas of organocatalysis, asymmetric reactions and green solvents. Very similar work, albeit not concentrated only to organocatalysis, was published recently (https://doi.org/10.3390/molecules27196701). The manuscript is organized according to used solvents, which on one hand gives a clear idea of the scope but on the other hand, some comparison concerning the same reaction type in different solvents is missing. Not only the comparison between the possible bio-based alternatives, but also some comparison with reactions in conventional solvents would be appropriate. In many cases presented here it can be assumed that the particular bio-based solvent was not used as a substitute for some other, less "green" solvent, but as the most effective solvent for the studied reaction regardless its bio origin (it especially applies for ethanol); this fact should be mentioned where applicable. In the present form, the review is merely a list of tested reactions giving no evalution, general rules or future prospects; this should be improved (the same applies for the conclusion). I appreciate the unified graphics of schemes, helpful color coding and inclusion of mechanism (though in my opinion the label "transition state" is inappropriate, in fact, the key step of reaction mechanism, not the transition state itself, is depicted). Unfortunately, the manuscript is rather "crude" and both the text and schemes contain a lot of errors (given below). Also the language is not very good, even poor in places, obscuring the meaning of the text (wrong word order or even missing subject or predicate, wrong sequence of tenses, prepositions, spelling mistakes etc.). As a whole, the manuscript can potentially be a very useful piece of work but it needs a considerable improvement.
Questions which should be addressed in the text:
What kind of reactions and catalysts is applicable in which solvent and why? Do they have something in common?
Is bio-based solvent always the best choice? Which factors have to be taken into account? What about problems with solubility? Above all, the solvent has to dissolve reaction components.
What about scCO2? It can be also considered a green alternative.
Other remarks:
- The use of abbreviations is not unified in the text (et al. x and coll. x e coll.; Naph x Naphth; EtOAc x AcOEt);
- it is more common to speak about reaction conditionS, as a complex of more conditions is meant;
- Scheme 2: the structure in transition state (TS) does not correspond to the structure of catalyst (different number of carbons); the same applies for Scheme 4;
- Scheme 3: EtCOOCH2COPh is not an activated methylene, I suppose it should be EtOCOCH2COPh, the same applies for PhCH2COOCH2COCH3; the enamine structure in TS does not correspond with any of the substrates 10a-h;
- Scheme 9 is mostly the same as Scheme 8 including TS, the only addition is the second step, which is in fact not an asymmetric catalytic reaction, so its mentioning in the text would be enough; ciclo(alkyl) instead of cycloalkyl (also applies to Schemes 37 and 46);
- Scheme 10: what does it mean GP? It should probably mean PG (protecting group) but it is not possible to revert it like that (and it would be appropriate to define PG at the first use or to list it the same way as R substituents); the same applies to Scheme 41;
- Scheme 12: missing H+ on the right side of the TS reaction – the overall charge is not equal;
- Scheme 15: what esther is? Ether or ester? The same applies for Scheme 16;
- lines 349-350: compounds 64 and 65 are not malonates but malononitriles;
- the reaction in Scheme 23 is a combination of H-bonding and covalent activation and this fact should have been mentioned, also the similarities and differences between reactions in Schemes 23 and 20 (same substrates, slightly different catalysts);
- Scheme 24: overlapping (entangled) bonds in the structure of calixarene; the reaction itself can hardly be labelled as asymmetric due to results;
- lines 389-399: very poor English;
- the reactions in Schemes 26 and 27 use the same catalyst and analogous substrates and yields analogous products, yet a different TS is suggested and they have different names;
- Scheme 28: the structure in TS does not correspond to the structure of catalyst 80;
- line 423: what does it mean stereoselective approximation? Should it be a stereosel. approach?
- Scheme 29: TS are marked as 81a,b instead of 82a,b. What does Wa and Wb mean? Identical yields and ee for both catalysts; is it correct?
- Scheme 30: what is CHOOR2 substituent in 85b? In TS, the structure of the substrate (benzaldehyde) is not consistent with substrate 1, the other substrate (should be a malonate) is bearing peroxo groups;
- Scheme 31: product 89c does not correspond to the general structure of product 89, moreover it is not chiral. Is it explained in the original paper?
- Scheme 32: the structure in TS is not a lactate;
- Scheme 34: general structures of 94 and 97 are not consistent with the structures of product examples;
- line 514 and Scheme 35: in case of selected examples, the diastereoselectivity can be hardly labelled as good, it is moderate at best;
- Scheme 37: missing X in the general structure of 104, R1, R2 should be R2, R3;
- Scheme 41: products 115b,d,e do not have defined stereochemistry;
- lines 601-603 do not make sense. Should not due to be in place of despite?
- line 627: the reference to Scheme 44 should be moved to more appropriate part of text, as the reaction mentioned in the last sentence is not included in it;
- lines 632-639: NHC is the catalytically active species and triazolium 122 is a precatalyst! This reaction is commonly known as oxidative esterification or (in case of a,b-unsaturated aldehydes, where no additional oxidant is needed) redox esterification. The whole paragraph is rather confused. The misunderstanding regarding the role of NHC repeats also in the subsequent paragraph (line 644) and both schemes (45, 46). Compound 126 is nor a catalyst, not NHC, but a precatalyst;
- line 638: a more common abbreviation for this base is DIPEA;
- Scheme 47: at least the general structure of product 130 should be given in full, without using an abbreviation Ns, which is not as commonly known as Ts;
- lines 664-665: according to Scheme 48, it is not possible to label the reagents 131 as p-tolylsulfenyl chlorides;
- line 677: according to Scheme 49, the substrate is not a carbonate but a carboxylate – which one is correct, text or scheme?
- Scheme 49: a part of the catalyst molecule is missing in TS;
- Scheme 51: the general structures of substrate 142 and product 145 is confusing, it would be better to give 2 structures – one for cyclic compounds and the other for acyclic a-fluoroketones;
- line 711: according to Scheme 52, 149f should be 148b;
- line 712: please specify the substrate used for the reaction at a gram scale;
- line 764: not only N-methyl substituted substrates 15 are included in Scheme 57;
- Scheme 59: substrate 167 is identical with 10a (ethyl acetoacetate);
- line 805: substrate 174 has an incorrect name (also in the caption to Scheme 61). Possibly it should have been a‑(……sulfonyl) ketones?
- Scheme 60: incorrect structure of products – one carbon is missing;
- Scheme 61: various substituents R in TS are not distinguished;
- Scheme 63: not all newly formed bonds are marked in black (3 bonds are formed in the reaction sequence); in Schemes 64-65 this color coding of new bonds is completely absent;
- lines 861-865: the whole sentence needs some rearrangement, it is confusing and suggests that DCM was necessary for the whole sequence, not only for the second step (which is not an asymmetric organocatalysis, so the used solvent is irrelevant with respect to the topic of the manuscript). The same applies for the next paragraph;
- Scheme 67: COOMe group in substrate and TS is changing to COOEt in the intermediate 191.
Author Response
The review by Martelli et al. is dealing with an important topic, a superposition of popular areas of organocatalysis, asymmetric reactions and green solvents. Very similar work, albeit not concentrated only to organocatalysis, was published recently (https://doi.org/10.3390/molecules27196701).
Indeed, the review “Application of Biobased Solvents in Asymmetric Catalysis” by Miele et al. includes examples of asymmetric oragnocatalysis, as well as metal and biocatalysis. This publication was cited in the reviewed version of our manuscript.
The manuscript is organized according to used solvents, which on one hand gives a clear idea of the scope but on the other hand, some comparison concerning the same reaction type in different solvents is missing. Not only the comparison between the possible bio-based alternatives, but also some comparison with reactions in conventional solvents would be appropriate. In many cases presented here it can be assumed that the particular bio-based solvent was not used as a substitute for some other, less "green" solvent, but as the most effective solvent for the studied reaction regardless its bio origin (it especially applies for ethanol); this fact should be mentioned where applicable. In the present form, the review is merely a list of tested reactions giving no evalution, general rules or future prospects; this should be improved (the same applies for the conclusion).
OK
I appreciate the unified graphics of schemes, helpful color coding and inclusion of mechanism (though in my opinion the label "transition state" is inappropriate, in fact, the key step of reaction mechanism, not the transition state itself, is depicted).
Thank you.
Unfortunately, the manuscript is rather "crude" and both the text and schemes contain a lot of errors (given below). Also the language is not very good, even poor in places, obscuring the meaning of the text (wrong word order or even missing subject or predicate, wrong sequence of tenses, prepositions, spelling mistakes etc.). As a whole, the manuscript can potentially be a very useful piece of work but it needs a considerable improvement.
The manuscript was carefully reviewed.
Questions which should be addressed in the text:
What kind of reactions and catalysts is applicable in which solvent and why? Do they have something in common? Is bio-based solvent always the best choice? Which factors have to be taken into account? What about problems with solubility? Above all, the solvent has to dissolve reaction components.
In the examples cited, we have included the solvent screening and comments on their choices.
What about scCO2? It can be also considered a green alternative.
We have included the following examples with scCO2:
Green asymmetric synthesis of tetrahydroquinolines in carbon dioxide medium promoted by lipophilic bifunctional tertiary amine - squaramide organocatalysts, 10.1016/j.tet.2017.11.057
Asymmetric catalytic synthesis of functionalized tetrahydroquinolines in supercritical fluids, 10.1016/j.supflu.2015.11.004
Stereodivergent Michael addition of diphenylphosphite to alpha-nitroalkenes in the presence of squaramide-derived tertiary amines: an enantioselective organocatalytic reaction in supercritical carbon dioxide, 10.1039/c3gc41647j
Supramolecular Assemblies of Amphiphilic L-Proline Regulated by Compressed CO2 as a Recyclable Organocatalyst for the Asymmetric Aldol Reaction, 10.1002/anie.201302662
Enantioselective addition of carbon acids to alpha-nitroalkenes: the first asymmetric aminocatalytic reaction in liquefied carbon dioxide, 10.1016/j.tetlet.2012.04.123
Other remarks:
- The use of abbreviations is not unified in the text (et al. x and coll. x e coll.;)
( Naph x Naphth; EtOAc x AcOEt);
OK
- it is more common to speak about reaction conditionS, as a complex of more conditions is meant;
OK
- Scheme 2: the structure in transition state (TS) does not correspond to the structure of catalyst (different number of carbons); the same applies for Scheme 4;
OK
- Scheme 3: EtCOOCH2COPh is not an activated methylene, I suppose it should be EtOCOCH2COPh, the same applies for PhCH2COOCH2COCH3; the enamine structure in TS does not correspond with any of the substrates 10a-h;
all corrections have been made
- Scheme 9 is mostly the same as Scheme 8 including TS, the only addition is the second step, which is in fact not an asymmetric catalytic reaction, so its mentioning in the text would be enough; ciclo(alkyl) instead of cycloalkyl (also applies to Schemes 37 and 46);
OK
- Scheme 10: what does it mean GP? It should probably mean PG (protecting group) but it is not possible to revert it like that (and it would be appropriate to define PG at the first use or to list it the same way as R substituents); the same applies to Scheme 41;
OK
- Scheme 12: missing H+ on the right side of the TS reaction – the overall charge is not equal;
OK
- Scheme 15: what esther is? Ether or ester? The same applies for Scheme 16;
This has been fixed
- lines 349-350: compounds 64 and 65 are not malonates but malononitriles;
OK
- the reaction in Scheme 23 is a combination of H-bonding and covalent activation and this fact should have been mentioned, also the similarities and differences between reactions in Schemes 23 and 20 (same substrates, slightly different catalysts);
All the facts mentioned were inserted in the text
- Scheme 24: overlapping (entangled) bonds in the structure of calixarene; the reaction itself can hardly be labelled as asymmetric due to results;
This example has removed from the text.
- lines 389-399: very poor English;
OK
- the reactions in Schemes 26 and 27 use the same catalyst and analogous substrates and yields analogous products, yet a different TS is suggested and they have different names;
OK
- Scheme 28: the structure in TS does not correspond to the structure of catalyst 80;
The structure of the organocatalyst has been corrected
- line 423: what does it mean stereoselective approximation? Should it be a stereosel. approach?
OK
- Scheme 29: TS are marked as 81a,b instead of 82a,b. What does Wa and Wb mean?
This has been fixed
Identical yields and ee for both catalysts; is it correct?
It’s correct
- Scheme 30: what is CHOOR2 substituent in 85b? In TS, the structure of the substrate (benzaldehyde) is not consistent with substrate 1, the other substrate (should be a malonate) is bearing peroxo groups;
These issues have been fixed.
- Scheme 31: product 89c does not correspond to the general structure of product 89, moreover it is not chiral. Is it explained in the original paper?
The product 89c does not correspond to the general structure of product 89, by reason of the carbonyl substrate has underwent primary nucleophilic attack of the N-center. The racemization of the product is not explained by the authors.
- Scheme 32: the structure in TS is not a lactate;
This has been fixed
- Scheme 34: general structures of 94 and 97 are not consistent with the structures of product examples;
Structures have been fixed.
- line 514 and Scheme 35: in case of selected examples, the diastereoselectivity can be hardly labelled as good, it is moderate at best;
OK
- Scheme 37: missing X in the general structure of 104, R1, R2 should be R2, R3;
This has been fixed
- Scheme 41: products 115b,d,e do not have defined stereochemistry;
This has been fixed
- lines 601-603 do not make sense. Should not due to be in place of despite?
OK
- line 627: the reference to Scheme 44 should be moved to more appropriate part of text, as the reaction mentioned in the last sentence is not included in it;
OK
- lines 632-639: NHC is the catalytically active species and triazolium 122 is a precatalyst! This reaction is commonly known as oxidative esterification or (in case of a,b-unsaturated aldehydes, where no additional oxidant is needed) redox esterification. The whole paragraph is rather confused. The misunderstanding regarding the role of NHC repeats also in the subsequent paragraph (line 644) and both schemes (45, 46). Compound 126 is nor a catalyst, not NHC, but a precatalyst;
OK
- line 638: a more common abbreviation for this base is DIPEA;
OK
- Scheme 47: at least the general structure of product 130 should be given in full, without using an abbreviation Ns, which is not as commonly known as Ts;
OK
- lines 664-665: according to Scheme 48, it is not possible to label the reagents 131 as p-tolylsulfenyl chlorides;
OK
- line 677: according to Scheme 49, the substrate is not a carbonate but a carboxylate – which one is correct, text or scheme?
OK
- Scheme 49: a part of the catalyst molecule is missing in TS;
OK
- Scheme 51: the general structures of substrate 142 and product 145 is confusing, it would be better to give 2 structures – one for cyclic compounds and the other for acyclic a-fluoroketones; OK
- line 711: according to Scheme 52, 149f should be 148b;
This has been fixed
- line 712: please specify the substrate used for the reaction at a gram scale;
OK
- line 764: not only N-methyl substituted substrates 15 are included in Scheme 57;
OK
- Scheme 59: substrate 167 is identical with 10a (ethyl acetoacetate);
OK
- line 805: substrate 174 has an incorrect name (also in the caption to Scheme 61). Possibly it should have been a‑(……sulfonyl) ketones?
OK
- Scheme 60: incorrect structure of products – one carbon is missing;
This has been fixed
- Scheme 61: various substituents R in TS are not distinguished;
This has been fixed
- Scheme 63: not all newly formed bonds are marked in black (3 bonds are formed in the reaction sequence); in Schemes 64-65 this color coding of new bonds is completely absent;
OK
- lines 861-865: the whole sentence needs some rearrangement, it is confusing and suggests that DCM was necessary for the whole sequence, not only for the second step (which is not an asymmetric organocatalysis, so the used solvent is irrelevant with respect to the topic of the manuscript). The same applies for the next paragraph;
OK
- Scheme 67: COOMe group in substrate and TS is changing to COOEt in the intermediate 191.
OK
Round 2
Reviewer 3 Report
The authors fixed most of the factical problems and the quality of the manuscript was significantly improved, however, despite some spell-checking was done, many problems with English remain (wrong syntax or sequence of tenses, singular x plural mismatch, inappropriate words - eg. the method obtained three examples... instead of yielded or afforded on line 426, founded instead of found on line 560 etc.). I strongly recommend reviewing the manuscript by a native speaker to further improve the quality of language.
Moreover, I still miss more thorough authors' assessment of the results as a whole and general conclusion regarding the topic. My questions (repeated below) remained mostly unaddressed.
What kind of reactions and catalysts is applicable in which solvent and why? Do they have something in common? Is bio-based solvent always the best choice? Which factors have to be taken into account? What about problems with solubility? Above all, the solvent has to dissolve reaction components.
Other remarks:
- Scheme 9: GP instead of PG in the mechanism
- Scheme 11: there is still a charge disbalance between both sides of the equilibrium in the mechanism; I suppose a proton is missing.
- Scheme 14: allyl is mentioned as R2, but most depicted products contain vinyl: which is correct, the structures or the name?
- lines 380+381: malonitrile (malic acid derivative) is not the same compound as malononitrile (malonic acid derivative)! There are malononitrile derivatives (64 and 65) in the reaction
- lines 421-432 (formerly 389-399): no improvement of English compared to previous version, multiple syntax and other errors)
- Scheme 27: What does Wa and Wb mean? Shall it be 80a and 80b?
- Scheme 29: if the formerly discussed product 89c (its numbering was not corrected to 87c, by the way) is a result of a different kind of transformation, then it should be either explained in the text, or the product omitted from the scheme/exchanged for any other of the 22 mentioned alternatives
- Scheme 35: ciclo(alkyl) should be corrected
- Scheme 39: GP instead of PG in the mechanism
- line 698: …obtained in situ by a deprotonation of precatalyst… would be more appropriate
- Scheme 43 caption: …catalyzed by NHC derived from triazolium 120 would be more appropriate; the term pre-NHC is not used
- line 709: as above - …use of triazolium 124 as a NHC catalyst precursor… would be more appropriate than …use of pre-NHC 124 as a precatalyst…
- Scheme 44 caption: …catalyzed by NHC derived from triazolium 124 would be more appropriate
- Scheme 44: ciclo(alkyl) should be corrected
- line 745: substrate 134 is not a carbonate but a carboxylate
- line 771: oxindole 142 should be oxindole 140
- line 851: not only N-methyl substituted isatins 15 are included in Scheme 57!
- Scheme 63 caption: thiourea derivative… would be more appropriate than …derived thiourea…
- Scheme 65: please include pressure and temperature values to reaction conditions
- Scheme 66 caption: …thiourea derivative… would be more appropriate than …thiourea-derived…
- Scheme 72: as the N-H hydrogen also comes from the substrate 204, it should be also labeled by violet color
Author Response
Once again, we would like to thank the constructive criticisms of the reviewer. All the corrections and additions are highlighted in the revised version of the manuscript, and the comments and answers are given in sequence, as follows:
The authors fixed most of the factical problems and the quality of the manuscript was significantly improved, however, despite some spell-checking was done, many problems with English remain (wrong syntax or sequence of tenses, singular x plural mismatch, inappropriate words -
- the method obtainedthree examples... instead of yielded or afforded on line 426
Actually, it was on line 446: OK
founded instead of found on line 560 etc.).
OK
I strongly recommend reviewing the manuscript by a native speaker to further improve the quality of language.
Moreover, I still miss more thorough authors' assessment of the results as a whole and general conclusion regarding the topic. My questions (repeated below) remained mostly unaddressed.
We have included further comments in the conclusions.
What kind of reactions and catalysts is applicable in which solvent and why? Do they have something in common?
Unfortunately, it is not possible to generalize since the structure of substrates may change completely the reactivity.
Is bio-based solvent always the best choice?
Not always the best yields and/or enantioselectivity are obtained with bio-based solvents, however the environmental impact of toxic solvents such as the chlorinated should be taken in to account. In special, at the pharmaceutical industry, more sustainable processes are clearly required.
Which factors have to be taken into account? What about problems with solubility? Above all, the solvent has to dissolve reaction components.
As we explain before, small changes in substrate structure may completely modify solubility and thus reactivity, thus it is quite difficult to generalize the applicability of bio-based solvents on asymmetric organocatalysis. However, as we have demonstrated in this review, a large number of well succeed examples has already been reported in literature, thus stimulating other research groups to at least evaluate these alternative greener solvents in the future.
Other remarks:
- Scheme 9: GP instead of PG in the mechanism
OK
- Scheme 11: there is still a charge disbalance between both sides of the equilibrium in the mechanism; I suppose a proton is missing.
OK
- Scheme 14: allyl is mentioned as R2, but most depicted products contain vinyl: which is correct, the structures or the name?
OK, these structures have been corrected.
- lines 380+381: malonitrile (malic acid derivative) is not the same compound as malononitrile (malonic acid derivative)! There are malononitrile derivatives (64 and 65) in the reaction.
OK, these terms have been corrected.
- lines 421-432 (formerly 389-399): no improvement of English compared to previous version, multiple syntax and other errors)
OK
- Scheme 27: What does Wa and Wb mean? Shall it be 80a and 80b?
OK
- Scheme 29: if the formerly discussed product 89c (its numbering was not corrected to 87c, by the way) is a result of a different kind of transformation, then it should be either explained in the text, or the product omitted from the scheme/exchanged for any other of the 22 mentioned alternatives
OK
- Scheme 35: ciclo(alkyl) should be corrected
OK
- Scheme 39: GP instead of PG in the mechanism
OK
- line 698: …obtained in situ by a deprotonation of precatalyst… would be more appropriate
ok
- Scheme 43 caption: …catalyzed by NHC derived from triazolium 120 would be more appropriate; the term pre-NHC is not used
ok
- line 709: as above - …use of triazolium 124 as a NHC catalyst precursor… would be more appropriate than …use of pre-NHC 124 as a precatalyst…
ok
- Scheme 44 caption: …catalyzed by NHC derived from triazolium 124 would be more appropriate
ok
- Scheme 44: ciclo(alkyl) should be corrected
ok
- line 745: substrate 134 is not a carbonate but a carboxylate
ok
- line 771: oxindole 142 should be oxindole 140
ok
- line 851: not only N-methyl substituted isatins 15 are included in Scheme 57!
ok
- Scheme 63 caption: thiourea derivative… would be more appropriate than …derived thiourea…
ok
- Scheme 65: please include pressure and temperature values to reaction conditions.
The values have been added to the scheme.
- Scheme 66 caption: …thiourea derivative… would be more appropriate than …thiourea-derived…
ok
- Scheme 72: as the N-H hydrogen also comes from the substrate 204, it should be also labeled by violet color.
The color has been changed.